# Lymphocyte-specific protein 1 regulates mechanosensory oscillation of podosomes and actin isoform-based actomyosin symmetry breaking

Pasquale Cervero [1], Christiane Wiesner[1], Anais Bouissou[2], Renaud Poincloux [2] & Stefan Linder [1]

Subcellular fine-tuning of the actomyosin cytoskeleton is a prerequisite for polarized cell migration. We identify LSP (lymphocyte-specific protein) 1 as a critical regulator of acto-myosin contractility in primary macrophages. LSP1 regulates adhesion and migration, including the parameters cell area and speed, and also podosome turnover, oscillation and protrusive force. LSP1 recruits myosin IIA and its regulators, including myosin light chain kinase and calmodulin, and competes with supervillin, a myosin hyperactivator, for myosin regulators, and for actin isoforms, notably β-actin. Actin isoforms are anisotropically distributed in myosin IIA-expressing macrophages, and contribute to the differential recruitment of LSP1 and supervillin, thus enabling an actomyosin symmetry break, analogous to the situation in cells expressing two myosin II isoforms. Collectively, these results show that the cellular pattern of actin isoforms builds the basis for the differential distribution of two actomyosin machineries with distinct properties, leading to the establishment of discrete zones of actomyosin contractility.

[1] Institut für Medizinische Mikrobiologie, Virologie und Hygiene, Universitätsklinikum Eppendorf, Martinistr. 52, 20246 Hamburg, Germany. [2] Institut de Pharmacologie et Biologie Structurale, IPBS, CNRS, UPS, Université de Toulouse, 205 route de Narbonne, 31077, Toulouse BP64182, France. Correspondence and requests for materials should be addressed to S.L. (email: s.linder@uke.de)

Macrophages constitute a crucial part of the innate immune system and are involved in counteracting infections and maintaining tissue homeostasis[1]. The ability of macrophages to migrate and to invade the extracellular matrix (ECM)[2] is based on their adaptable morphology[3], and the local degradation of matrix components[4]. Both functions are regulated by the actin cytoskeleton, especially by actomyosin-based contractility. To induce polarized migration, a break in cellular symmetry, especially in the pattern of actomyosin activity, is necessary. This can include differential recruitment of myosin isoforms, such as myosin IIA and IIB[5] or local relaxation of the actomyosin cortex[6]. However, as macrophages express predominantly myosin IIA[7], the respective mechanism is unclear.

A symmetry break in macrophages involves reorganization of the actin cytoskeleton, notably the recruitment of podosomes to the leading edge. Podosomes constitute prominent actomyosin-based organelles of the cell cortex, in monocytic cells such as macrophages[8], immature dendritic cells[9] and osteoclasts[10], and also in endothelial[11], smooth muscle[12] and neural crest cells[13]. Podosomes feature an extensive repertoire of functions such as cell–matrix adhesion, extracellular matrix degradation, topography and rigidity sensing, and others, which makes them crucial regulators of macrophage migration and invasion[14].

Podosomes contain an F-actin-rich core, surrounded by a ring of adhesion plaque proteins such as talin[15] or vinculin[16]. Both substructures are anchored to the ECM by transmembrane proteins such as CD44[17] and integrins[18]. Unbranched lateral actin filaments surround the podosome core[19], while a second set of unbranched actin filaments connects podosomes into higher-ordered clusters[19,20]. Recent research points to the existence of a cap structure on top of the podosome[14]. Identified cap components comprise the formins FMNL1[21] and INF2[22], and also supervillin[20], a member of the villin family. Supervillin forms a hub for actoymyosin[23] at the cell cortex, by binding directly to myosin IIA and actin through regions within its N-terminal half[23,24], and to myosin regulators such as the long form of myosin light chain kinase (L-MLCK)[25]. Supervillin is a myosin IIA hyperactivator, as it binds activated myosin and also induces activation, leading to a feed-forward cycle and to podosome dissolution[20].

We now identify leukocyte-specific protein 1 (LSP1) as a myosin IIA-associated regulator of macrophage migration and invasion, and a novel component of the podosome cap. LSP1 is recognized as a regulator of immune cell migration in inflammation and phagocytosis[26,27], with aberrant LSP1 overexpression in neutrophil actin dysfunction (NAD47/89) leading to reduced motility of neutrophils and severe recurrent infections[28–31], and LSP1 deficiency leading to enhanced T cell migration, contributing to the development of rheumatoid arthritis[32]. However, LSP1´s molecular modes of action, and its interplay with other regulators of the actomyosin cortex are unclear. We now show that LSP1 interacts with actin, myosin IIA, and specific regulators of myosin activity, including L-MLCK and calmodulin. Importantly, LSP1 competes with supervillin for binding of these regulators in cells, leading to the formation of distinct zones of myosin contractility.

We further show that differential recruitment of LSP1 and supervillin correlates with the subcelluar patterning of actin isoforms. Mammalian cells can express several of up to six actin isoforms that are grouped into three clusters, comprising α-skeletal muscle, α-smooth muscle and α-cardiac actin, β-cytoplasmic actin, as well as γ-smooth muscle and γ-cytoplasmic actin[33,34], with the α/β/γ isoform designation based on variant electrophoretic mobility, due to the number and type of acidic residues in their N-termini[35]. Studies from knock out mice indicated that, despite overlapping functions,

actin isoforms can not fully compensate for each other[33,34]. In consequence, impairment of specific isoforms can lead to pathologies, such as hearing loss, based on compromised stereocilia maintenance in the case of γ-cytoplasmic actin[36]. Actin isoform function has been speculated to involve differential binding of specific interaction partners, such as cofilin[37] or profilin[38]. Moreover, actin isoforms were shown to be differentially distributed, for example β- and γ-cytoplasmic actin in fibroblasts and endothelial cells[39]. However, the validity of both concepts is under discussion.

Our data provide a molecular explanation for the reported effects of LSP1 in immune cell dysregulation. We also show how competitive binding for actin isoforms and myosin regulators between LSP1 and supervillin, two actomyosin modulators with different activity, can lead to an actomyosin symmetry break and enable polarized migration of immune cells.

## Results

**LSP1 is enriched at podosomes and the macrophage leading edge.** A proteomic screen of podosome-enriched fractions of primary human macrophages pointed to LSP1 as a potential new podosome component (Supplementary Fig. 1A)[40]. This was confirmed by staining of endogenous LSP1, F-actin and vinculin (Fig. 1a–d). Z-sections showed that LSP1 forms a cap structure on top of the core that extends along the sides of the podosome, thus partially overlapping with both core and ring (Fig. 1a′–d′), depending on the plane of imaging (Fig. 1d′, e). To analyse LSP1 localization at higher resolution, macrophages were stained for LSP1 and F-actin and analysed by STED (stimulated emission depletion) microscopy (Supplementary Fig. 1B–D). Z-sections (Supplementary Fig. 1E) and 3D reconstructions (Fig. 1f) confirmed that LSP1 is enriched at the podosome cap and also at lateral fibers.

Macrophages harbour two subpopulations of podosomes, larger and highly dynamic precursors at the cell periphery and the leading edge[41], and smaller successors at the inner region of cells[20]. Consistent with its general enrichment at the cell periphery, LSP1 localized mostly at precursors, and to a lesser degree at successors in migrating cells (Supplementary Fig. 1F–J). Analysis of the respective LSP1/F-actin intensity confirmed that LSP1 is especially enriched in a 5–10 μm wide zone at the cell periphery in both resting and migratory macrophages (Supplementary Fig. 1K–M). This preferential enrichment at the cell periphery was also observed in live cell analysis of resting macrophages coexpressing GFP-LSP1 and lifeact-RFP (Supplementary Fig. 2A–L). Interestingly, GFP-LSP1 and lifeact-RFP followed similar dynamics during formation and dissolution of single podosomes (Supplementary Fig. 2M–O).

To identify the domains of LSP1 that determine its subcellular localization, GFP-fused deletion constructs were created, comprising constructs containing only the N-terminal half ("N-terminal"), only the C-terminal half ("C-terminal"), only the caldesmon-like domains ("C1C2"), and only the villin headpiece-like domains ("V1V2") (Fig. 1g). Confocal analysis (Supplementary Fig. 3A–O) showed that the C-terminal half of LSP1 (Supplementary Fig. 3G–I), and especially the villin headpiece-like domains (Supplementary Fig. 3M–O), determine LSP1's localization to podosomes and to the actin cortex.

**LSP1 depletion enhances dynamics of podosomes and cells.** SiRNA-targeting of LSP1 led to depletion of 48 and 56% for two individual sequences (Supplementary Fig. 4A). Side-by-side stainings of cells treated with LSP1-specific or control siRNA showed that these reductions were also visible on the single cell-level, while the ability for podosome formation was not impaired

(Supplementary Fig. 4B–G). Podosome dynamics were analysed by confocal live cell imaging of cells expressing lifeact-RFP. Each frame of a respective video (1 h duration) was color coded for time, moving structures thus appear in multiple colors, while static ones are white. Accordingly, stationary macrophages treated with control siRNA showed mostly podosomes in white, while migrating cells showed progressive coloration of podosome clusters from the trailing to the leading edge, consistent with dissolution of podosomes in the back, and formation of new podosomes at the front (Fig. 2a). Strikingly, cells treated with

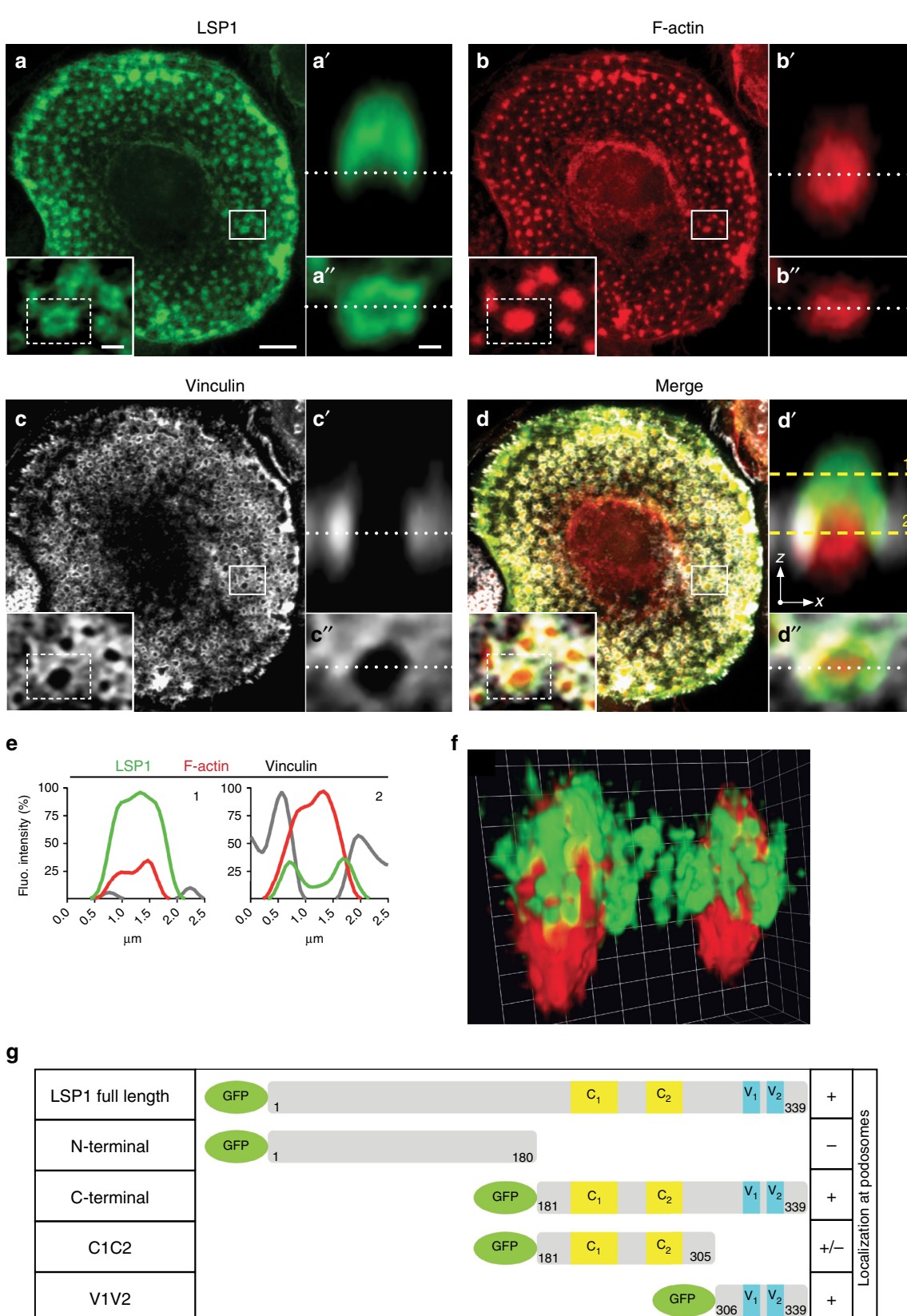

LSP1-specific siRNA showed strongly enhanced dynamics of podosome clusters (Fig. 2b, c). Podosome clusters, tracked by their centers of mass, were mostly static or showed persistent trajectories in controls (Fig. 2d), whereas clusters in cells treated with LSP1 siRNA were highly motile and showed non-directional trajectories (Fig. 2e, f), accompanied by an increased velocity of clusters in cells treated with LSP1 siRNA (0.87 ± 0.12 μm/min and 0.73 ± 0.1 μm/min), corresponding to a relative increase of 281 and 236% (Fig. 2g).

Live cell analyses showed that podosomes in LSP1 siRNA-treated cells have reduced lifetimes (6.6 ± 0.5 min and 6.6 ± 0.3 min; "±" indicating s.e.m.), compared to controls (9.9 ± 0.6 min), corresponding to a reduction of 33% (Fig. 2h), which was based on a general shift to shorter life times (Fig. 2i). Of note, macrophages transfected with LSP1-specific siRNA often showed longer trajectories (Fig. 2j–l), based on an increase (38 and 46%) of cell speed (0.33 ± 0.05 μm/min and 0.35 ± 0.05 μm/min for LSP-1 siRNA vs. 0.24 ± 0.02 μm/min for controls) (Fig. 2m), while net displacement of cells was not significantly altered. Interestingly, LSP1 knockdown also led to a pronounced increase (53 and 90%) in cell area (2226 ± 209 μm² and 2771 ± 307 μm² for LSP1 siRNA vs. 1456 ± 83 μm² for controls), pointing to a role of LSP1 in cortex stabilization and/or contractility (Fig. 2n).

To further evaluate changes in podosome distribution upon LSP1 knockdown, cells were scored in groups, according to evenly spaced distribution of podosomes ("uniform"), recruitment of podosomes to a single leading edge ("polarized") and formation of clusters ("clusters"). In line with live cell imaging observations (Fig. 2a–c), LSP1 knockdown cells showed prominent formation of podosome clusters (40.2 ± 6.3% and 48.2 ± 5.2% for LSP1 siRNAs vs. 19.1 ± 7.1% for controls) (Fig. 2o). To assess changes in overall cell morphology, cells were evaluated for circularity, a measure for equidistance of all points on the cell perimeter to the center, and also for aspect ratio, a measure for cell elongation, by determining the ratio of longest versus shortest axis[42] (see Materials). Frequency distribution analysis showed that aspect ratio was enhanced in LSP1 knockdown cells, with ~40–50% of cells showing values between 1.5 and 2.5, compared to ~20% of controls (Fig. 2p), while circularity was strongly decreased, with ~20% of LSP1 knockdown cells showing values between 0.85 and 0.95, compared to ~60% of controls (Fig. 2q). Scatter plots of both values (Fig. 2r) further showed that only ~20% of control cells displayed extreme values, for aspect ratio above 1.3, and for circularity below 0.8 (Fig. 2s), in contrast to ~60% of LSP1 knockdown cells (Fig. 2t–w). Collectively, these data indicate that LSP1 knockdown macrophages were more elongated and less circular. Moreover, they also showed an increased tendency to deviate from the inverse correlation between aspect ratio and circularity observed in control cells (Fig. 2s–u), indicating an uncoupling of localized cell protrusion and overall cell polarization.

To investigate subcellular localization of LSP1 in a 3D context, macrophages expressing LSP1-GFP and Lifeact-RFP were embedded in 3D collagen gels and analysed by confocal microscopy. LSP1-GFP was strongly enriched in F-actin-rich protrusions (Supplementary Fig. 5A–C), pointing to a potential role of LSP1 also in macrophage invasion. To test this, invasion was analysed in a collagen plug invasion assay[4] (Supplementary Fig. 5D–F). The invaded area was comparable for cells treated with LSP1-specific and control siRNA (Supplementary Fig. 5H). However, the number of invading cells was increased in case of LSP1 depletion (166±13% for siRNA1, 150±4% for siRNA2, compared to controls; Supplementary Fig. 5G), showing that LSP1 also has a regulatory role in 3D invasion of macrophages.

**LSP1 regulates oscillatory protrusion of podosomes.** Podosomes are contractile organelles, with growth of the Arp2/3 complex-generated actin core[43] exerting forces on the formin-based lateral actin cables[22,44], and mechanical coupling of both systems enabling protrusion into the matrix[45–48]. To analyse the impact of LSP1 on podosome protrusion, F-actin levels at podosomes were measured in a confocal plane over time. As reported for dendritic cells[49], Lifeact-RFP-based fluorescence of individual podosomes varied in an oscillatory fashion, indicating podosome movement in the Z axis (Fig. 3a, a′, c). Overexpression of LSP1-GFP (Fig. 3b, b′, d) led to an increased number (by 49%) of local peaks (4.3 ± 0.2 for LSP1-GFP cells, vs. 2.9 ± 0.1 for controls) (Fig. 3e), while peak height was only slightly reduced (by 15%) (17.4 ± 1.0 for LSP1-GFP cells, vs. 20.5 ± 1.0 for controls) (Fig. 3f), and the frequency distribution of peak height was slightly shifted to smaller values (Fig. 3g). In contrast, depletion of LSP1 (Fig. 3i, i′, k) led only to a slight decrease (by 17%) in the number of local peaks (2.6 ± 0.2 for LSP1 siRNA cells, vs. 3.1 ± 0.1 for controls) (Fig. 3l), while peak height was strongly increased (by 53%) (32.8 ± 2.3 for LSP1 siRNA cells, vs. 21.4 ± 1.3 for controls) (Fig. 3m), and the frequency distribution of peak height showed broader deviation from control values, towards higher maximal values, indicative of a pronounced irregularity in peak height (Fig. 3n).

To measure the forces podosomes impose on the matrix, protrusion force microscopy was used. This technique applies atomic force microscopy to cells that are seeded on pliable membrane sheets (Fig. 3o)[48]. Protrusion force was determined for macrophages treated 4 days previously with LSP1-specific siRNA or control siRNA. Cells treated with LSP1 siRNA showed a 32% reduction of the evaluated protrusion force (1.7 ± 0.2 nN), compared to controls (2.5 ± 0.2 nN) (Fig. 3p). Collectively, these results show that LSP1 is a crucial regulator of mechanosensory oscillation of podosomes, ensuring both the regularity of this process and also the generation of protrusive force.

**LSP1 recruits myosin IIA to the cell cortex and to podosomes.** The influence of LSP1 on podosome dynamics and function, both of which are based on actomyosin contractility[48–50], pointed

**Fig. 1** LSP1 is a component of the podosome cap structure. **a–d** Confocal micrographs of a macrophage stained for LSP1 using specific primary antibody and Alexa 488-labeled secondary antibody (**a**, green), for F-actin using Alexa405-labeled phalloidin (**b**, red), and for vinculin using specific primary antibody and Alexa568-labeled secondary antibody (**c**, white), with merge (**d**). White boxes in **a–d** indicate detail images shown as insets. Dashed boxes in insets indicate single podosome shown in x–z section in **a′–d′** and in x–y section in **a″–d″** with respective cross-section planes shown as white dotted lines. Dashed yellow lines in **d′** indicate confocal planes used for measurements of respective fluorescence intensities shown in **e**. Note the cap-like localization of LSP1 on top of the F-actin core (1), which can appear ring-like in a lower optical section (2) Scale bars: 5 μm in **a–d**, 1 μm in insets, 0.5 μm in **a″–d″**. **f** 3D reconstruction of two podosomes from STED micrographs macrophages stained for LSP1 (green) and F-actin (red), scale unit: 0.25 μm. See also Supplementary Movie 1. **g** Domain structure of LSP1 full length and deletion mutants. LSP1 features an acidic N-terminal half containing a hypothetical Ca²⁺ binding domain, two caldesmon-like F-actin binding domains (C1, C2) and two villin-headpiece-like F-actin binding domains (V1, V2). First and last amino acid residues are indicated. "+" and "−" indicate the presence or absence of the respective construct at podosomes

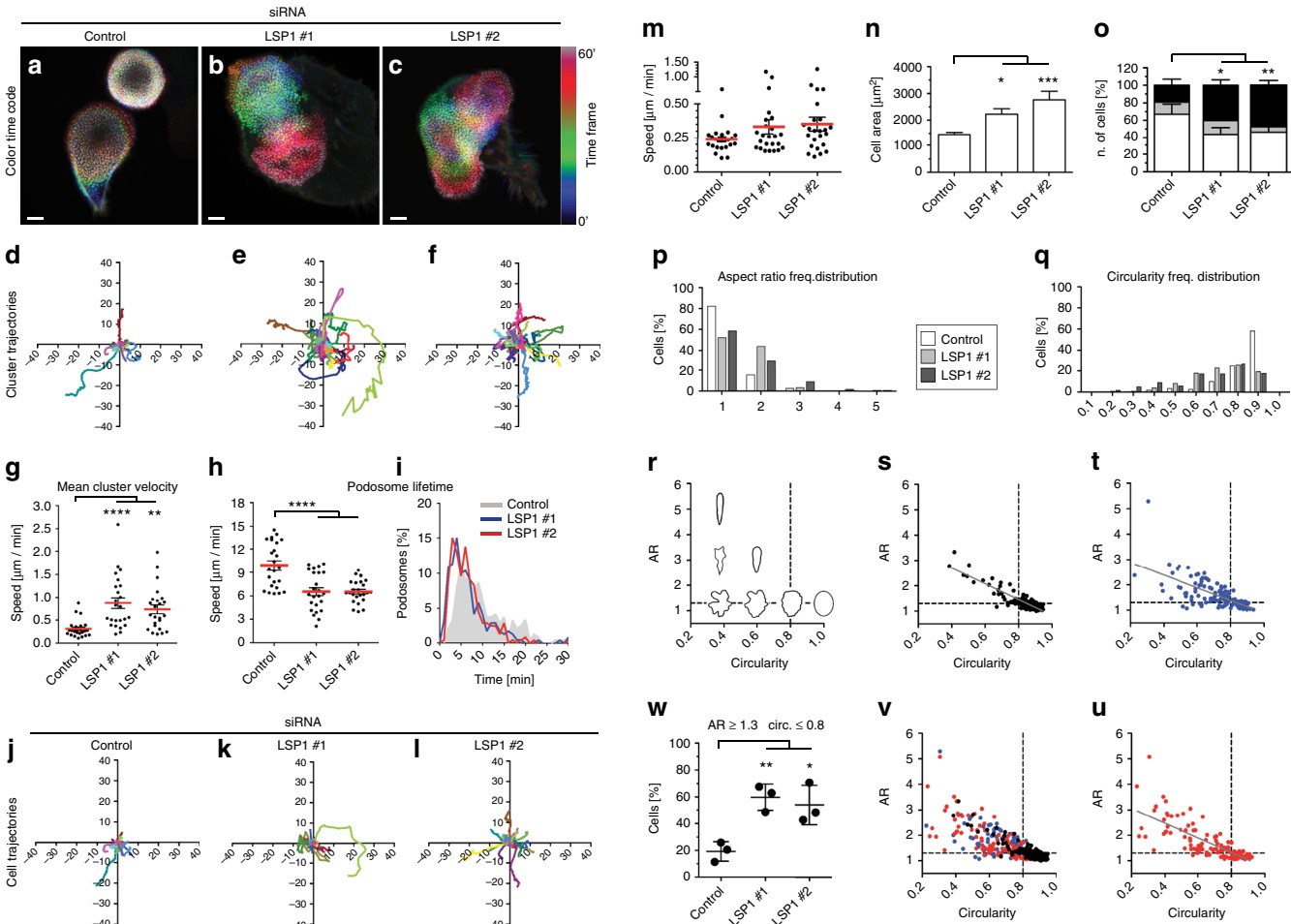

**Fig. 2** LSP1 depletion leads to enhanced dynamics of podosomes, podosome clusters and cells. **a–c** Still images from confocal time lapse videos, frames were progressively colored along the spectrum, with final merges presented. Scale bars: 10 μm. See Supplementary Movies 2–4. **d–f** Colored lines track centers of mass of podosome clusters from cells of four different donors. Axes indicate relative position in μm. **g** Velocity of podosome clusters (μm min$^{-1}$), in 24 cells from four different donors. Dots represent mean cluster velocity of single cells. **h, i** Podosome lifetime, with 240 podosomes evaluated per condition. Dots in **h** represent mean values of 10 podosomes in single cells. **i** Frequency distribution of podosome lifetime. **j–l** Coloured lines track centers of mass of cells from four different donors. Axes indicate relative position in μm. **m, n** Graphs show speed (μm min$^{-1}$) (**m**), or total area (μm$^2$) (**n**) of cells treated with indicated siRNAs. Dots represent mean value of single cells ($n = 24$), each bar represents mean value of 24 cells from four different donors. **o–w** Cell morphology in control and LSP1 knockdown cells. **o** Podosome distribution, with cells scored in groups, according to even distribution (white), recruitment of podosomes to single leading edge (grey) and formation of clusters (black). Values represent the percentage of at least 50 cells per donor ($N = 3$). **p, q** Frequency distribution for aspect ratio (**p**) or circularity (**q**), with 123 cells from three different donors evaluated. **r–v** Scatter plots of circularity (x-axes) and aspect ratio (y-axes), with schematic graph depicting respective cell shapes (**r**), and plots for cells treated with control siRNA (**s**), LSP1-specific siRNAs (**t, u**), and merge (**v**). Cut-off values of 1.3 for aspect ratio and 0.8 for circularity: dashed lines, regression line in grey. Dots represent single cells ($N = 123$) from three donors. (**w**) Percentage of cells showing aspect ratio ≥1.3 and circularity ≤0.8. Dots represent mean percentages calculated with at least 35 cells per donor ($N = 3$). **a–w** Treatment with specific siRNAs is indicated. Values are given as mean ± s.e.m. Statistical test: one-way ANOVA. *$P < 0.05$, **$P < 0.01$, ***$P < 0.001$, ****$P < 0.0001$. For specific values, see Supplementary Data 1

to a functional connection between LSP1 and myosin IIA, the predominant form of myosin in macrophages[7] and at podosomes[20,51]. Fluorescence intensity measurements of F-actin at podosome cores (Fig. 4b, d) and myosin IIA levels at regions surrounding and including podosomes cores (Fig. 4a, c) showed a reduction of myosin IIA, but not of F-actin levels, for LSP1 siRNA-treated cells (myosin IIA intensity of LSP1-siRNA treated cells: 31.8 ± 16.3 and 35.9 ± 11.9 %, vs. 54.1 ± 21.3 for controls) (Fig. 4e, f), with measured areas being comparable between treatments (Fig. 4g, h).

To test a potential interaction between LSP1 and myosin IIA, and to determine the respective subcellular sites, proximity ligation assays (PLA) were performed, which report close spatial proximity (<40 nm) between two proteins. Control cells showed

PLA background levels (2.4 ± 0.4 spots per cell; Fig. 4i–l), whereas cells stained with LSP1 and myosin IIA-specific antibodies showed a strong increase in PLA spots (73 ± 7.8 spots per cell; Fig. 4m–o, l), with respective areas being comparable (Fig. 4p). PLA signals were enriched at the cell periphery, where precursor podosomes are located[41], and at the cell cortex. Similar results were gained with a PLA analysis of LSP1 and pan-actin, indicating interaction of LSP1 and actin especially in the cell periphery (Supplementary Fig. 6A–H). STED microscopy (Supplementary Fig. 6K–N) confirmed the close spatial proximity of myosin IIA and LSP1 in the cell periphery. Of note, podosomes in this zone showed colocalizing pixels especially at their periphery-facing side (Supplementary Fig. 6M), and LSP1 and myosin IIA colocalized at discrete spots along the whole podosome cap

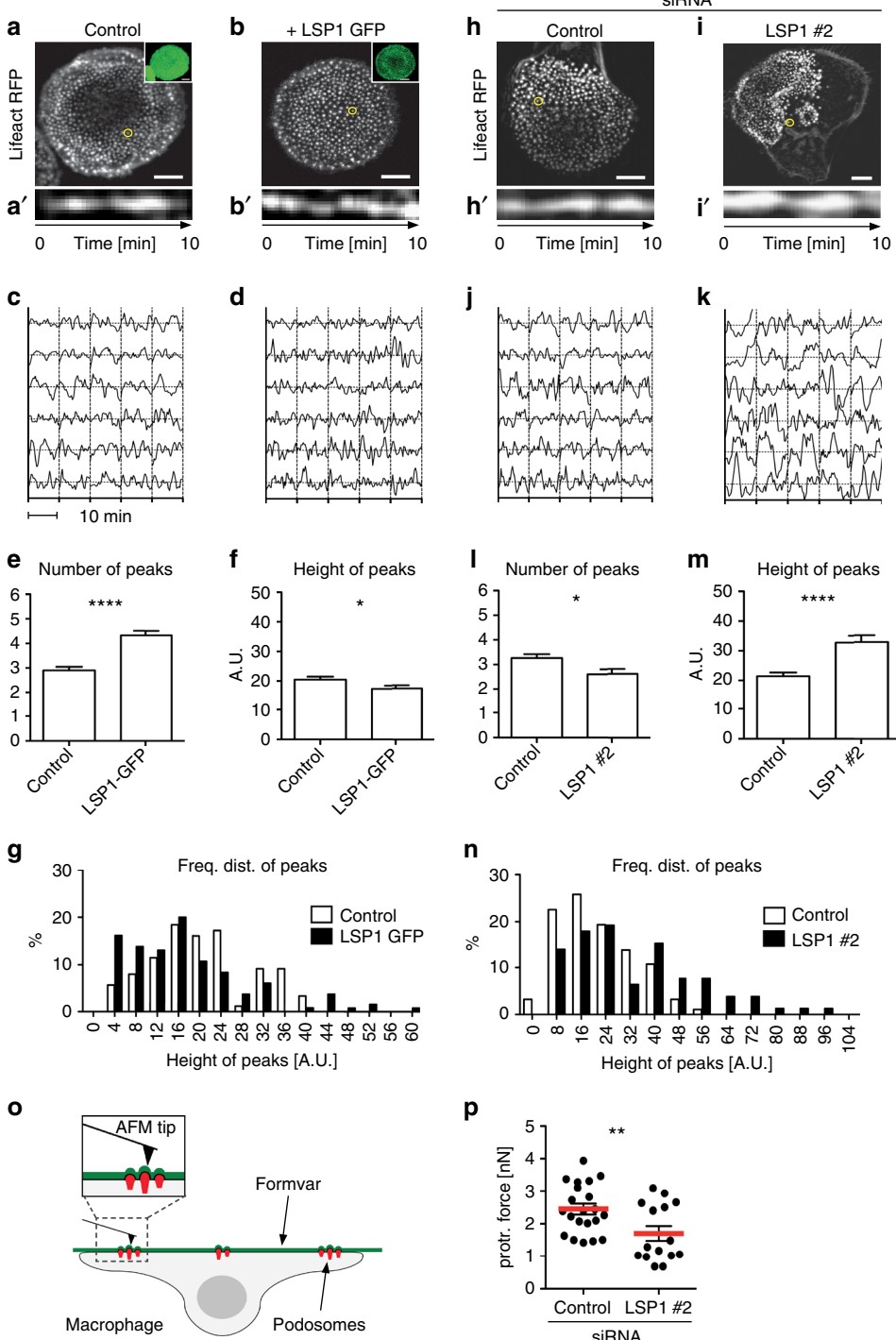

**Fig. 3** LSP1 is a regulator of oscillatory podosome protrusion. **a**, **b**, **h**, **i** Podosome oscillation, determined by F-actin intensity measurements. Confocal micrographs of cells expressing lifeact-RFP and coexpressing EGFP (**a**) or LSP1-GFP (**b**), the latter shown as small insets in green, or treated with control (**h**) or LSP1-specific siRNA (**i**). **a**′–**b**′, **h**′–**i**′ Kymographs of F-actin intensities at single podosomes indicated by yellow circles in **a**, **b**, **h**, **i**. Time is indicated in min. **c**, **d**, **j**, **k** Normalized F-actin intensities at single podosomes in a fixed plane of focus from EGFP control cells (**c**), LSP1-GFP expressing cells (**d**), or cells treated with control siRNA (**j**) or LSP1-specific siRNA (**k**). Each row shows F-actin intensity curves of five podosomes from a single cell, with 6 cells from from different donors analysed. Duration of each individual measurement: 10 min. **e**, **f**, **l**, **m** Statistical evaluation of F-actin based fluorescence intensity fluctuations, with **e**, **l** showing number of peaks and **f**, **m** showing height of peaks Data shown are Mean ± s.e.m.; *$P < 0.05$, ****$P < 0.0001$; two-tailed unpaired $t$-test. **g**, **n** Frequency distribution of peak height from graphs shown in **c**, **d** and **j**, **k**. Note different scale on $x$-axes due to broader frequency distributon in LSP1 kockdown cells. **o** Principle of protrusion force microscopy. A primary human macrophage is seeded on pliable Formvar matrix. Inversion of the setup allows probing of ventral macrophage surface, including oscillatory protruding podosomes, by atomic force microscopy (AFM). Modified from[48]. **p** Measurement of protrusion forces generated by single podosomes from cells treated with control or LSP1-specific siRNA. Each dot represents the mean protrusion force of at least 23 podosomes (max 245) in a single cell. Data collected from two donors. Values are given as mean ± s.e.m. **$P < 0.01$, two-tailed unpaired $t$-test. For specific values, see Supplementary Data 1

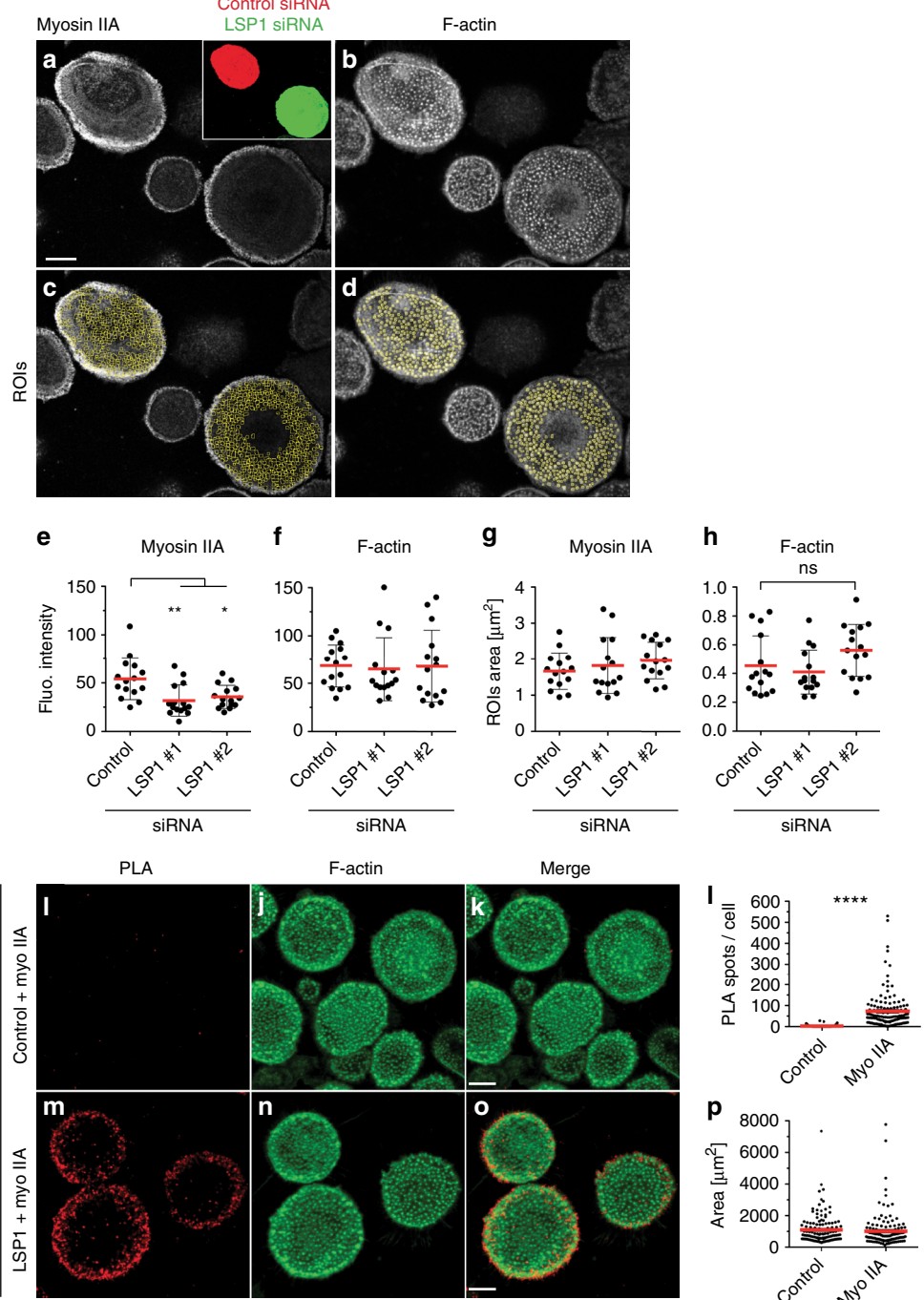

**Fig. 4** LSP1 interacts with myosin IIA and regulates its recruitment to podosomes. **a**, **b** Confocal micrographs of mixed populations of macrophages, stained for myosin IIA using specific primary antibody (**a**) or for F-actin using Alexa-405-labeled phalloidin (**b**). Cells treated with LSP1 siRNA express GFP, cells treated with control siRNA express mCherry, as shown in inset in **a**. **c**, **d** ImageJ-based macros were used to identify myosin IIA at podosomes (**c**) and F-actin-rich podosome cores (**d**). Scale bar: 10 μm. **e**–**h** Statistical evaluation of (**e**) myosin-based fluorescence at podosomes, (**f**) F-actin-based fluorescence at podosomes and size of areas analysed for podosomal myosin IIA (**g**) and F-actin (**h**) podosome-covered area. Each dot represents the mean intensity of all the individual podosomes detected in a single cell (~500 podosomes/cell on average), five cells from three different donors. Values are given as mean ± SD. *$P < 0.05$, **$P < 0.01$, one-way ANOVA test. **i**–**k**, **m**–**o** Confocal micrographs of macrophages subjected to a proximity ligation assay (PLA), using myosin IIA-specific antibody, together with control IgG (**i**) or LSP1-specific antibody (**m**) and stained for F-actin (**j**, **n**), with merges (**k**, **o**). Scale bars: 10 μm. Note low background in **i** and PLA signals, especially at the cell cortex, in **m**. **l**, **p** Statistical evaluation of number of PLA spots per cell (**l**) and cellular area analysed for respective PLAs (**p**). Each dot represents one cell. Data collected from two different donors. Values are given as mean ± s.e.m. ****$P < 0.0001$, two-tailed unpaired $t$-test. For specific values, see Supplementary Data 1

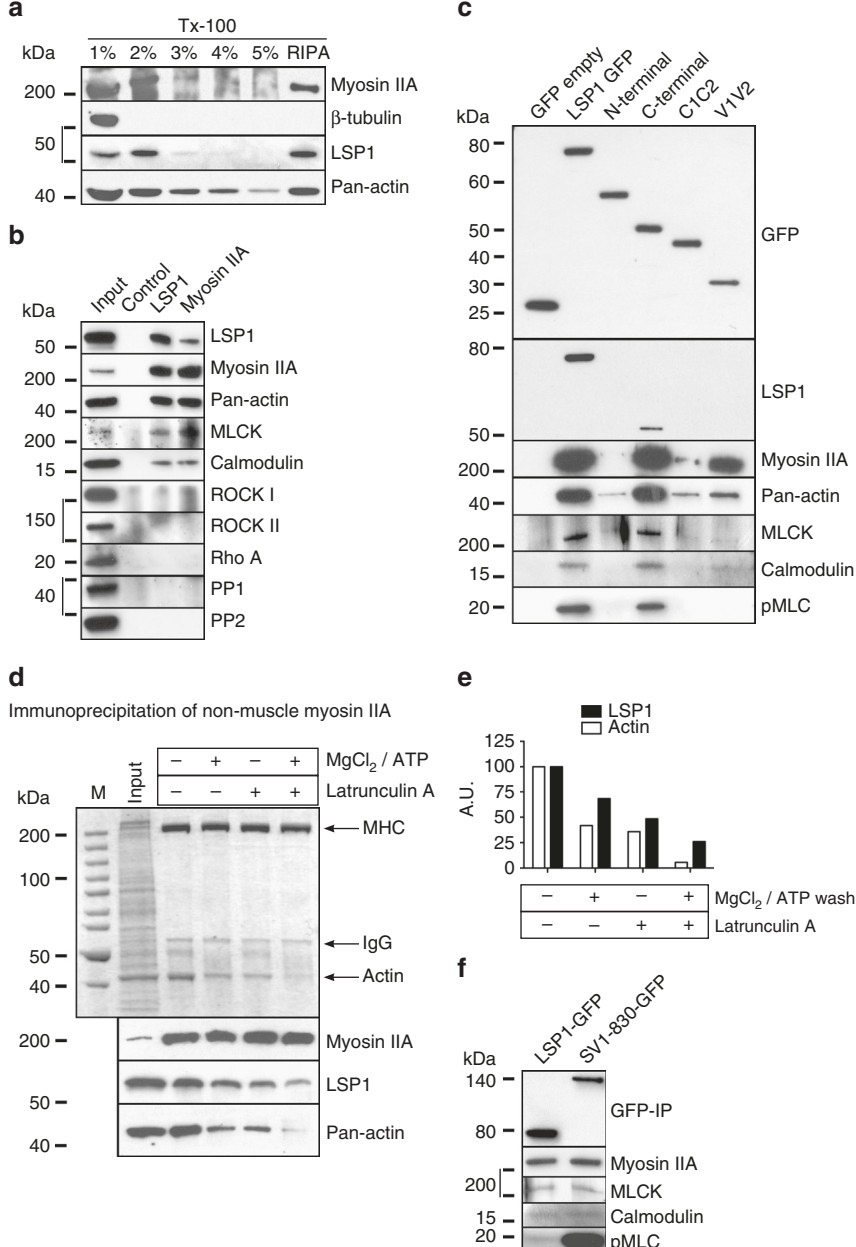

**Fig. 5** LSP1 and supervillin interact with a similar subset of myosin IIA regulators. **a** Western blots of macrophage lysates progressively extracted by increased concentration of TritonX-100 (1–5%), or in RIPA buffer. Note that LSP1, and also myosin IIA, are mostly extracted in the 2% and RIPA fractions. **b** Western blot of immunoprecipitation of endogenous LSP1 or myosin IIA from macrophage lysates, with control IgG and input. Blots were probed with indicated antibodies. Note cross-coprecipitation of LSP1 and myosin IIA, accompanied by coprecipitation of myosin regulators MLCK and calmodulin. Molecular weight in kDa is indicated. **c** Western blots of anti-GFP immunoprecipitation of lysates from macrophages expressing LSP1 full length or LSP1 domains fused to GFP (For inputs, see Supplementary Fig. 7A). Blots were probed with indicated antibodies. Note coprecipitation of myosin IIA with full length LSP1, and also with the C-terminal and the villin headpiece-like domains (V1V2) of LSP1. Interestingly only the full length and the C-terminal constructs, but not V1V2, are also able to bind MLCK, pMLC and calmodulin. An LSP1-positive band in the lane of the C-terminal construct probably reflects the fact that the anti-LSP1 antibody recognizes an epitope in the C-terminal half of LSP1. **d** Immunoprecipitation of myosin IIA from macrophage lysates, with addition of $Mg^{2+}$/ATP and/or latrunculin A (10 μM), as indicated. Upper panel: colloidal Coomassie stained SDS PAGE gel, lower panels: corresponding western blots developed with indicated antibodies. Molecular weight is indicated in kDa. **e** Quantification of coprecipitated amounts of actin and LSP1, normalized to precipitations performed without addition of $Mg^{2+}$/ATP and/or latrunculin A. **f** Western blots of anti-GFP immunoprecipitations from lysates of macrophages expressing LSP1-GFP or supervillin construct SV1-830-GFP developed with indicated primary antibodies. Molecular weight in kDa is indicated. Note that LSP1 and SV1-830 coprecipitate comparable amounts of myosin IIA, MLCK and calmodulin, but that supervillin-coprecipitated myosin is more activated, as indicated by pMLC signal

(Supplementary Fig. 6N). The preferential association of LSP1 with the actin cortex was substantiated by detergent extraction of cell lysates, with the majority of the cellular LSP1 pool being present in the detergent-resistant fraction, which also contained actin and myosin IIA (Fig. 5a).

**The LSP1 C-terminus recruits myosin IIA and its regulators**. Confirming the PLA analysis (Fig. 4m–o), LSP1 and myosin IIA were cross-immunoprecipitated from macrophage lysates (Fig. 5b). This also resulted in the co-precipitation of a subset of myosin regulators, including the long form of myosin light

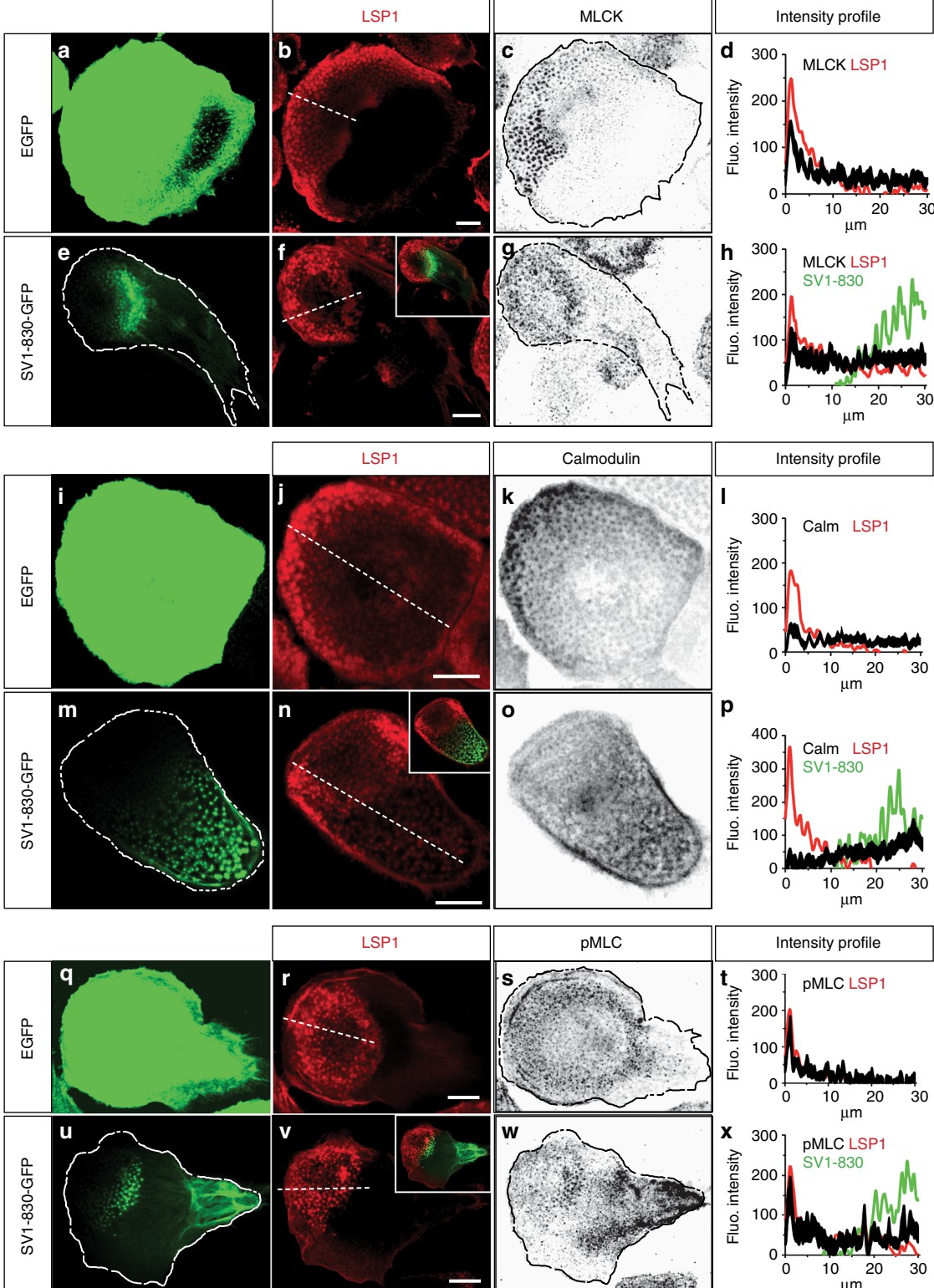

**Fig. 6** LSP1 and supervillin compete for myosin regulators in cells. Confocal micrographs of control cells expressing EGFP (**a**, **i**, **q**) or SV1-830-GFP (**e**, **m**, **u**), and stained for endogenous LSP1 **b**, **f**, **j**, **n**, **r**,**v** both stained for MLCK (**c**, **g**), calmodulin (**k**, **o**) or pMLC (**s**, **w**). Dashed white lines in **b**, **f**, **j**, **n**, **r**, **v** indicate pixels used for generation of intensity profiles shown in **d**, **h**, **l**, **p**, **t**, **x**, gained from analysis of 10 cells from three different donors. Graphs for LSP1 and supervillin show respective mean intensities, while graphs for MLCK, calmodulin and pMLC also include error bars. Note localization of SV1-830 to the trailing edges of cells, accompanied by redistribution of MLCK (**d**, **h**), calmodulin (**l**, **p**), and pMLC (**t**, **x**) to the rear of respective cells, and also to rearward-facing podosomes. Scale bars: 10 μm

chain kinase (L-MLCK) and calmodulin, but not of RhoA, Rho kinase (ROCK) 1 and 2 or protein phosphatase (PP) 1 and 2 (Fig. 5b), pointing to the potential existence of a complex consisting of LSP1, myosin IIA, as well as L-MLCK and caldmodulin.

To dissect the interaction sites of LSP1 with its partners in more detail, GFP-fused full length and truncation constructs of LSP1 (Fig. 1g), were expressed in macrophages and precipitated by anti-GFP immunoprecipitation. The C-terminal construct coprecipitated actin, myosin IIA, MLCK, phospho-MLC and calmodulin, comparable to full length LSP1 (Fig. 5c). The N-terminal and C1C2 constructs did not lead to significant co-precipitation of the probed LSP1 interaction partners, whereas the V1V2 construct retained partial binding to actin, myosin IIA and calmodulin (Fig. 5c; Supplementary Fig. 7A).

Immunoprecipitations of myosin IIA from macrophage lysates showed that depletion of actin from myosin-containing precipitates using $Mg^{2+}$/ATP and/or latrunculin A was accompanied by a reduction of coprecipitated LSP1 (Fig. 5d, e). In order to test whether the binding of myosin IIA by LSP1 is direct or indirect, myosin cosedimentation experiments using purified rabbit skeletal muscle myosin were performed. As expected, myosin formed filaments in vitro that were pelleted by ultracentrifugation[23]. However, neither full length nor C-terminal constructs of LSP1 were found to cosediment to a significant degree (Supplementary Fig. 7B). Collectively, these results suggest an indirect interaction of LSP1 and myosin IIA that is apparently mediated by F-actin.

**LSP1 and supervillin recruit the same myosin regulators**. The data pointed to LSP1 being in a complex with actin, myosin IIA and its regulators at the cell cortex and at podosomes. Of note, supervillin has been described as a similar regulator of myosin IIA activity both at podosomes[20], and at the cell cortex[25], by recruiting actin, myosin IIA and L-MLCK through its N-terminal region (SV 1-830)[20,23]. To investigate whether LSP1 and supervillin bind the same subset of myosin regulators, macrophages were transfected with LSP1-GFP or SV1-830-GFP, and anti-GFP immunoprecipitations were performed. The amount of coprecipitated myosin IIA was ×1.6 higher for supervillin, compared to LSP1. However, the level of myosin light chain phosphorylated at the Ser19 residue of pMLC was even more enhanced (×9.0), leading to a ×5.6 increase in the overall level of myosin activation, thus confirming the role of supervillin as a myosin hyperactivating protein[20] (Fig. 5f). In addition, supervillin was found to interact with L-MLCK, as reported[25], as well as calmodulin (Fig. 5f).

LSP1 and supervillin present as, respectively, moderate or strong regulators of myosin II activity. Next, we explored the consequences of their binding to a similar subset of myosin regulators in cells. Macrophages were transfected with SV1-830-GFP or GFP as controls, stained for endogenous LSP1 and co-stained for MLCK, calmodulin or pMLC. In both controls and SV1-830 expressing cells, LSP1 was localized to the leading edge of polarized cells (Fig. 6b, f, j, n, r, v). In contrast, expression of SV1-830 led to a redistribution of MLCK (Fig. 6c, d), calmodulin (Fig. 6k, l) and pMLC (Fig. 6s, t) from the leading edge and precursor podosomes towards the trailing edge of cells and successor podosomes (Fig. 6g, h, o, p, w, x), at sites of supervillin-driven myosin hyperactivation. (Note: based on Western blot densitometry and transfection rates, LSP1-GFP and SV1-830-GFP were calculated to be overexpressed by a factor of 6.5 and 11.0, respectively, compared to endogenous proteins, and by a factor of 1.6 relative to each other.)

**LSP1 and supervillin bind differentially to actin isoforms**. We next investigated the potential cause underlying this asymmetry, and in particular, whether differential binding and distribution of actin isoforms could be involved. Interestingly, α-cardiac actin was found to localize only at podosomes, and especially at the rearward part of podosome groups (Fig. 7a), where successor podosomes are found[20]. By contrast, β-actin was detected at cortical actin structures (Fig. 7b), including precursor podosomes[41], with γ-actin showing a similar distribution, however, with less enrichment at the cell cortex (Fig. 7c). (Note: α-cardiac actin was detected in the macrophage podosome proteome[40], in contrast to α- smooth muscle actin, which was also not detectable by Western blots or immunofluorescence in the present study). This could be corroborated by fluorescence intensity graphs of podosome fields from different cells ($n = 10$) (Fig. 7d, e) and by ratiometric analysis of α-cardiac actin over β- or γ-actin (Fig. 7f, g). This differential distribution of actin isoforms mirrors the localization of LSP1 and supervillin, with LSP1 localizing to the leading edge or periphery of cells (Fig. 6b, f, j, n, r, v), which are enriched in β- and γ-actin (Fig. 7b, c, i, l), and supervillin (GFP-SV1-830) localizing to the inner part of cells, where it closely follows the α-cardiac actin gradient (Fig. 7h, k), being especially enriched at sites of a high α-cardiac over β- or γ-actin ratio (Fig. 7j, m).

To further investigate the potential differences in actin isoform binding by LSP1 and supervillin, anti-GFP immunoprecipitates (Fig. 5d) were re-probed for α-cardiac, β-, and γ-actin (Fig. 7n). Densitometry showed that, while the LSP1 construct coprecipitated only ~2/3 of total actin, compared to SV1-830-GFP, the levels of coprecipitated α-cardiac actin were even lower, corresponding to a further reduction of ~50% (Fig. 7o). Altogether with the immunofluorescence data, these biochemical experiments thus pointed to a potential role of the α-cardiac-versus β-actin binding in the establishment of distinct subcellular zones that are respectively enriched in LSP1 or supervillin.

To test the potentially differential binding of LSP1 and supervillin to individual actin isoforms, LSP1 full length and the three actin binding regions of supervillin (SV171-342, SV 343-571, SV570-830)[23] were expressed in E. coli as 6xHis or GST fusions, respectively, purified and added to pure actin isoforms in F-actin cosedimentation assays, which are based on ultracentrifugation of polymerized actin. Controls showed that both β-actin from human platelets and bovine α-cardiac actin were almost completely pelleted, while 100% of LSP1 6xHis and >50% of the individual purified GST-fused F-actin binding fragments of supervillin remained in the supernatant when not mixed with pure actin isoforms (Fig. 8a; Supplementary Fig. 7C). Of note, addition of LSP1 6xHis to β-actin resulted in 84% of cosedimentation of the fusion protein, in contrast to only 45% cosedimentation with α-cardiac actin (Fig. 8a, b). Importantly, all supervillin GST constructs were found to coprecipitate to a comparable degree with either β- or α-cardiac actin (Supplementary Fig. 7C,D). Collectively, these results showed that LSP1 has a preference for the binding of β-actin, with a comparably ~50% reduced binding to α-cardiac actin, while supervillin F-actin binding regions exhibit no such preference for a specific actin isoform. Furthermore, live cell imaging showed that the preferential localization of LSP1-GFP to the β-actin-rich leading edge and of SV1-830 to the α-actin-rich inner regions of the cell, respectively, is very persistent, and even in a cell changing direction, respective subcellular zones are rapidly reestablished (Supplementary Fig. 8; Supplementary Movie 5).

To explore potential consequences of depletion of actin isoforms, we established siRNA-mediated knockdown of α-cardiac actin and β-actin. Treatment of macrophages with α-cardiac actin-specific siRNA led to a global ~1/3 reduction of α-

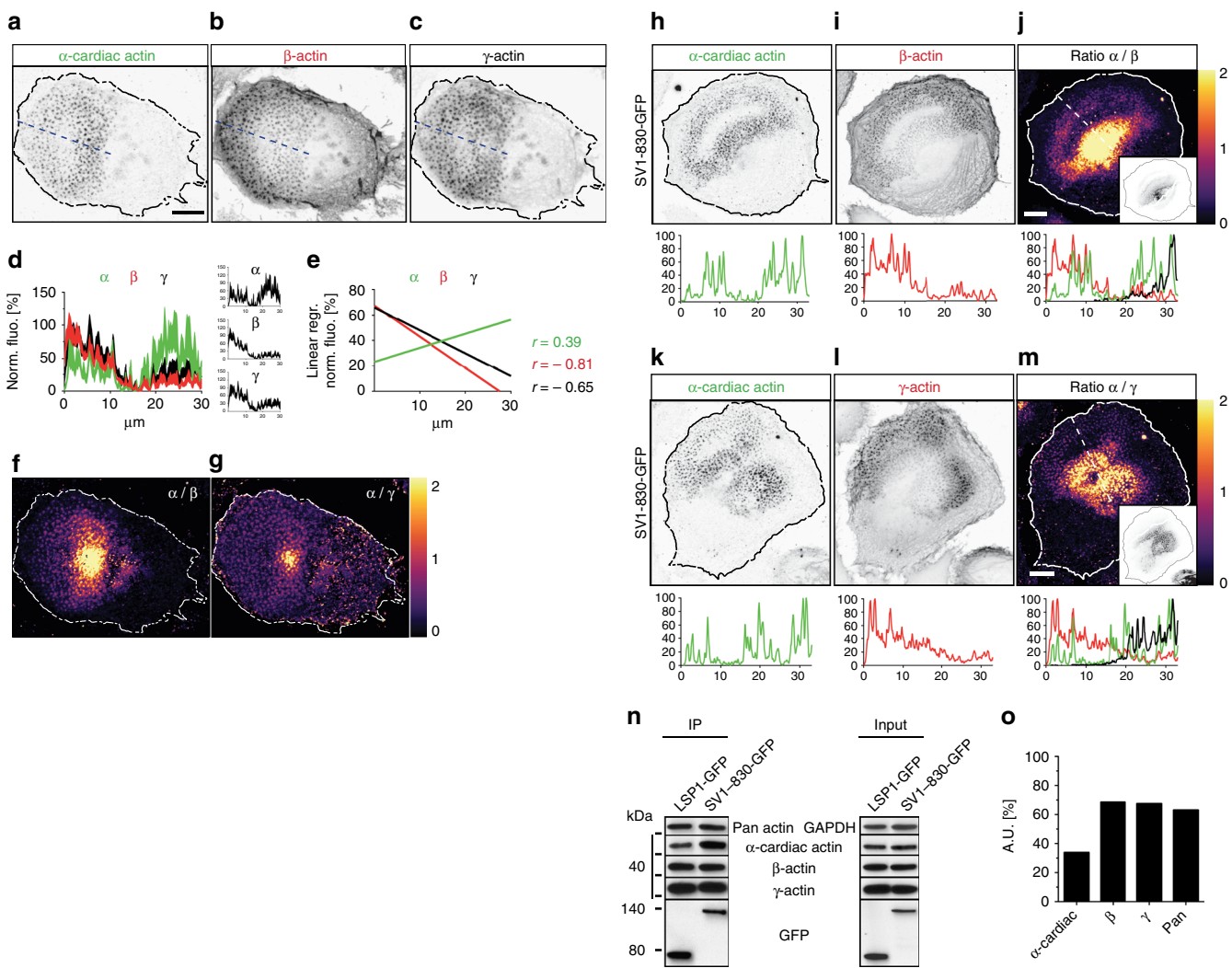

**Fig. 7** LSP1 and supervillin localize to different subcellular zones that are enriched in distinct actin isoforms. **a–c** Confocal micrographs of primary human macrophage, co-stained for endogenous α-cardiac (**a**), β- (**b**), and γ- (**c**) actin, shown in inverted grey scale. Bar: 10 μm. Dashed line indicates pixels used for respective fluorescence intensity graphs, with relative maximums set to 100%. **d** Fluorescence intensity diagram of α-cardiac, β- and γ-actin, average from 10 cells with comparable podosome-covered areas, with standard errors. Individual graphs are shown in diagrams on the right. **e** Linear regressions of graphs shown in (**d**) with respective Pearson correlation values (**r**). **f**, **g** Ratiometric analysis of stainings in **a–c**, showing enrichment of α-cardiac-actin over β-actin (**f**) or γ-actin (**g**). Fold enrichment is indicated by colour scale on the right. **h–m** Confocal micrographs of primary human macrophage overexpressing GFP-SV1-830 and co-stained for endogenous α-cardiac (**h**), and β-actin (**i**), or for α-cardiac (**k**) and γ-actin (**l**), shown in inverted grey scale. **j**, **m** Ratiometric analysis with fold enrichment indicated by colour scale on the right. Bar: 10 μm. Dashed lines indicates pixels used for respective fluorescence intensity graphs, with relative maximums set to 100%. Small insets show respective supervillin (SV 1-830-GFP) signals, with cell circumference indicated by black line. Note close correlation between α-cardiac actin enrichment and supervillin localization. **n** Western blots of anti-GFP immunoprecipitations (left panel) from lysates of macrophages expressing LSP1-GFP or supervillin construct SV1-830-GFP, with respective inputs (right panel), developed with indicated primary antibodies. Molecular weight in kDa is indicated. **o** Quantification of precipitated actin isoform bands from LSP1 GFP-IP lysates shown in **n** in comparison to SV1-830-GFP-IP set to 100%

cardiac actin levels (Supplementary Figure 9A, B), and a wide range of reduction on the individual cell level. Strikingly, cells depleted for α–cardiac actin, but still forming peripheral podosomes enriched in β-actin, and expressing SV1-830-GFP showed a shift of the supervillin construct towards the cell periphery (Supplementary Figure 9C–F). This was reminiscent of the phenotype of the isolated, myosin II-binding region of supervillin (SV1-174), which also localizes to unbranched, myosin IIA-enriched actin filaments at the cortex (Supplementary Figure 9G–J)[20]. Treatment with β-actin specific siRNA led to a reduction of ~ 60% of β-actin levels, again with varying effects on the individual cell level. Also, while global levels of the remaining actin isoforms, and also of LSP1, were unchanged in case of α-

cardiac actin or β-actin knockdown (Supplementary Figure 9B), individual β-actin depleted cells frequently also showed reduced staining for α-cardiac actin (Fig. 8g). Importantly, the peripheral localization of LSP1 was not discernibly affected as a result of β-actin knockdown (Fig. 8h), indicating that additional molecular mechanisms are involved in cortical localization of LSP1.

Combining these results, we reasoned that the preferred binding of β-actin by LSP1 could restrict the access of supervillin to the cell periphery and to precursor podosomes. To test this hypothesis, SV1-830 distribution was analysed in control (Fig. 9a–d) and LSP1-depleted cells (Fig. 9e–h). Indeed, the SV1-830 construct was no longer restricted to rearward podosomes in LSP1 knockdown cells, instead showing a much

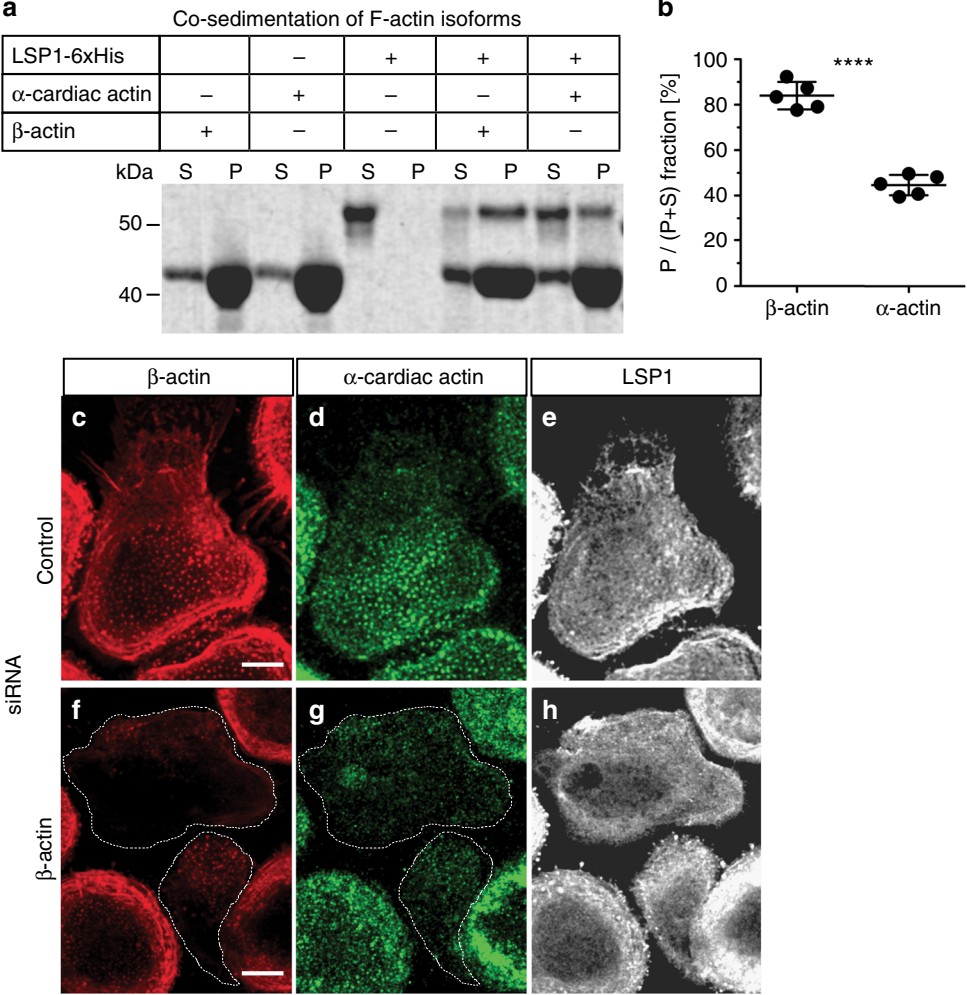

**Fig. 8** Differential subcellular recruitment of LSP1 and supervillin is based on the preferential binding of LSP1 to β-actin. **a** Comassie blue staining of gel showing actin cosedimentation assays with pure β-actin, α-cardiac actin, or LSP1 6xHis, alone as control, or in combination with actin isoforms, as indicated. Lanes showing supernatant and pellet fractions are labelled with "S" and "P", accordingly. Molecular weight is indicated in kDa. **b** Quantification of copelleted material as ratios of pelleted fraction versus input. Values are given as mean ± S.D.; ****$P < 0.0001$, two-tailed unpaired $t$-test. **c–h** Confocal micrographs of macrophages treated with control siRNA (**c–e**) or β-actin-specific siRNA (**f–h**), stained for β-actin (**c**, **f**), α-cardiac actin (**d**, **g**), and LSP1 (**e**, **h**). (Note: staining conditions required especially for α-cardiac actin are not optimal for staining of LSP1.) Scale bars: 10 μm. For specific values, see Supplementary Data 1

broader localization that extended towards the cell periphery (Fig. 9e–h). This was further corroborated by analysis of respective fluorescence intensities that showed a shift in the recruitment of SV1-830 towards the cell periphery in LSP1-depleted cells (Fig. 9i–l). Further analyses showed that SV1-830 overexpression led to a reduction of the morphological changes observed upon LSP1 knockdown. Accordingly, cell area of LSP1 depleted cells overexpressing SV1-830 (Fig. 9m) did not increase compared to control cells and to cells only depleted of LSP1 (Fig. 2n). Also, formation of podosome clusters was unchanged, compared to control cells (Fig. 9n), which was in contrast to LSP1 knockdown cells (Fig. 2m, n). Furthermore, both aspect ratio and circularity index were comparable to control values (Fig. 9o, p) and also showed a close correlation (Fig. 9q–t), in clear distinction to the deviation observed for LSP1 knockdown cells (Fig. 2s–v). Moreover, the number of cells showing more extreme values (AR ≥ 1.3; C ≤ 0.8) was reduced to ∼ 10–15%, in contrast to values of ∼ 60% for cells only treated with LSP1 siRNA, but without SV1-830 overexpression. Collectively, these results show that supervillin overexpression leads to a rescue of the morphological aberrancies observed upon LSP1 knockdown.

## Discussion

The data presented reveal new roles for two myosin IIA regulatory proteins in structural and sensory functions of primary macrophages that are crucial for migration and invasion of these cells. In particular, we identify LSP1 as a central regulator of the actomyosin cortex and show that LSP1 interacts with myosin IIA and its regulators, including L-MLCK, calmodulin and MLC. We also show that LSP1 competes with supervillin, a myosin hyper-activator[20], for these regulators, leading to discrete zones of myosin activity, and that establishment of these subcellular zones is based on the differential distribution of actin isoforms.

Consistent with earlier data[52–54], LSP1 was found to localize to the actin cortex and to podosomes. LSP1 regulates cell dynamics on multiple levels, as siRNA-induced knockdown of LSP1 led to enhanced dynamics of cells, podosome clusters and also of individual podosomes. LSP1 knockdown increased cell speed and migration track length, which is in line with leukocytes from LSP1-deficient mice showing faster migration to sites of inflammation[55], and overexpression of LSP1 in neutrophil actin dysfunction syndrome (NAD47/89) being based on reduced chemotactic motility of neutrophils[56]. LSP1 knockdown

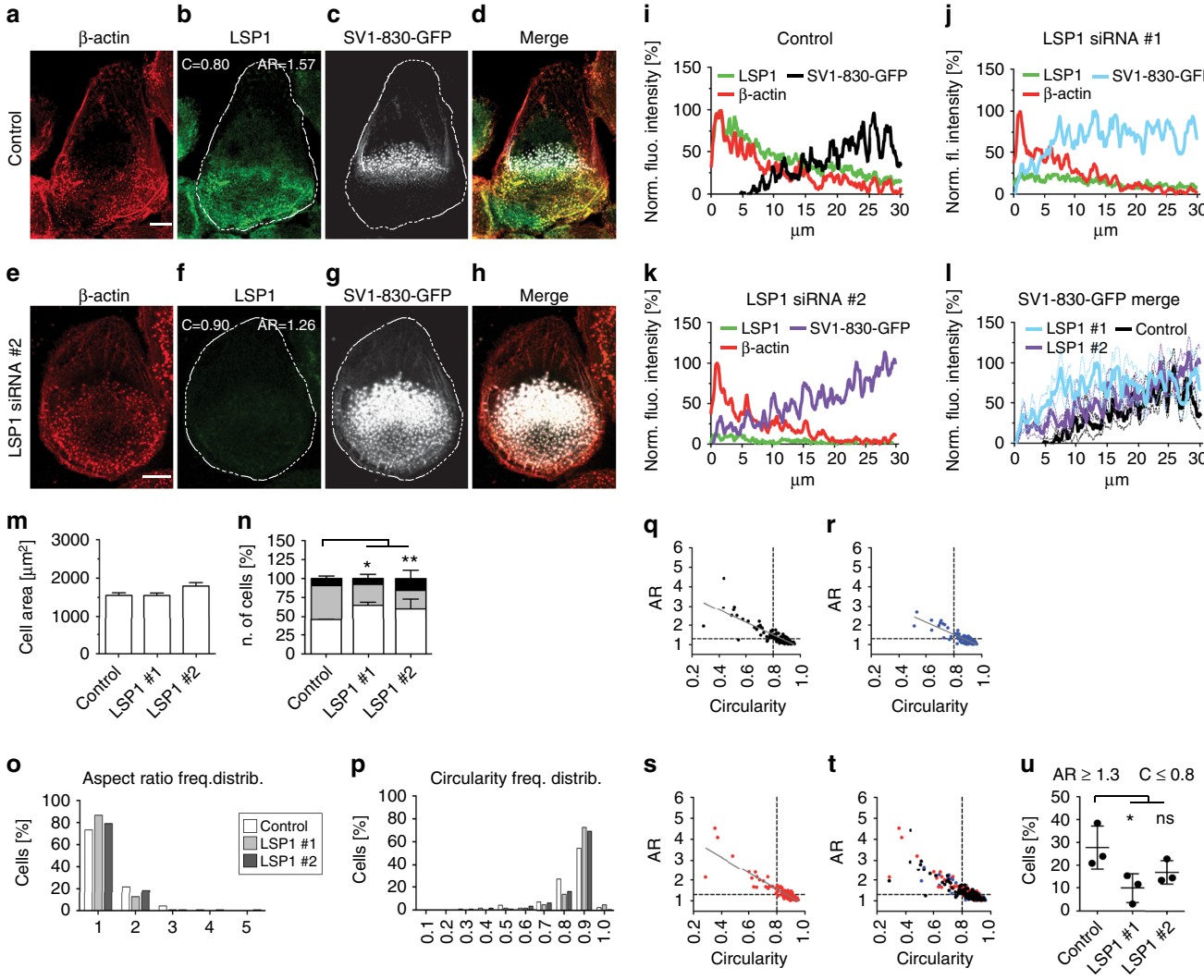

**Fig. 9** Supervillin overexpression rescues morphological aberrancies due to LSP1 depletion. **a–l** LSP1 blocks access of supervillin to β-actin-rich podosomes in the cell periphery. **a–h** Confocal micrographs of macrophages treated with control siRNA (**a–d**) or LSP1-specific siRNA (**e–h**) and overexpressing SV1-830-GFP, stained for β-actin (**a**, **e**), LSP1 (**b**, **f**), SV1-830-GFP signal (**c**, **g**), with merges (**d**, **h**). "C" and "AR" in (**b**, **f**) indicate circularity and aspect ratio. Scale bars: 10 μm. (**i–l**) Fluorescence intensity scans of β-actin, LSP1 and SV1-830-GFP in cells treated with control siRNA (**i**) or with two individual LSP1-specific siRNAs (**j**, **k**), or all SV1-830-GFP intensity scans shown together (**l**). Graphs show mean intensities from 30 μm long line scans taken from the cell periphery towards the rear in 10 cells from three donors. **m** Total area (μm²) of cells treated with indicated siRNAs and overexpressing SV1-830-GFP. Each bar represents mean value of 120 cells collected from three different donors. Values are given as Mean ± s.e.m. **n** Podosome distribution in cells treated with indicated siRNAs and overexpressing SV1-830-GFP. Cells were scored in groups, according to even distribution (white), recruitment of podosomes to a single leading edge (grey) and formation of clusters (black). Values are given as Mean ± S.D and represent the percentage calculated with at least 30 cells per donor (N = 3). *P < 0.05, **P < 0.01, one-way ANOVA test. **o–u** Evaluation of cell morphology in control and LSP1 knockdown cells that overexpress SV1-830-GFP. **o, p** Frequency distribution for cells grouped according to aspect ratio (**o**) or circularity (**p**) with 120 cells collected from three different donors evaluated. **q–t** Scatter plots of circularity (x-axes) and aspect ratio (y-axes), with plots for cells treated with control siRNA (**q**), LSP1-specific siRNAs (**r**, **s**), and merge (**t**). Cut-off values of 0.8 for circularity and 1.3 for aspect ratio are indicated by dashed lines. Regression line shown in grey. Each dot represents a single cell (N = 120) from three different donors. **u** Evaluation of cells expressing SV1-830-GFP, treated with indicated siRNAs and showing values for aspect ratio ≥1.3 and for circularity ≤0.8; Values are given as Mean ± S.D.,*P < 0.05, one-way ANOVA test. For specific values, see Supplementary Data 1

macrophages also showed an increase in cell size, consistent with LSP1 negatively regulating T-cell spreading[32], and pointing to relaxation of cortical tension as a possible result of LSP1 depletion. Conceivably, relaxation of cortical tension could also be involved in the enhanced 3D invasion of LSP1 knockdown macrophages, as it could facilitate squeezing of cells through ECM pores or formation of blebs to drive locomotion.

Using confocal and STED microscopy, we show that endogenous LSP1 localizes on top of the podosome core and partially extends along its side. Our data thus confirm and extend previous

observations of a cap-like structure on top of the podosome core. Remarkably, all of the identified components of the podosome cap, including FMNL1[21], INF2[22], and supervillin[20] are involved in the regulation of unbranched actin filaments and of myosin IIA, pointing to a function of the podosome cap in actomyosin-based contractility. This concept is supported by the role of LSP1 in podosome oscillation and protrusive force generation. Podosomes have been described as contractile organelles, in which growth of the core leads to exertion of forces on the lateral actin cables[44,45,47], resulting in oscillatory movement of podosomes in

the z direction that can be blocked by the myosin II inhibitor blebbistatin[22,49]. Our data show that LSP1-GFP overexpression leads to enhanced "frequency" of podosome oscillations, while LSP1 depletion results in a more irregular "amplitude". Furthermore, atomic force measurements showed that siRNA-induced depletion of LSP1 led to a ~30% reduction of podosome protrusive force.

As podosome oscillation is thought to be a central mechanism in detecting and reporting the rigidity of the extracellular matrix[45,46], LSP1 and the cap structure in general likely fulfill a critical function in mechanosensing, which would also tie in with the role of myosin IIA in this process[22,49], which we now identify as an interaction partner of LSP1. Of note, LSP1 binds to F-actin[57], but not to G-actin and also does not directly influence the rate of actin polymerization[31].

Endogenous LSP1 and myosin IIA were cross-precipitated in immunoprecipitations of macrophages lysates. Importantly, proximity ligation assays showed that both proteins closely colocalize at podosomes and at the cell cortex, compatible with a potential interaction. Similar results were gained for LSP1 and actin, pointing to the existence of a complex of LSP1, myosin IIA and actin. Myosin cosedimentation assays, and also myosin immunoprecipitation in the presence of latrunculin and $Mg^{2+}$/ATP, reducing the amount of coprecipitated actin filaments, showed that the interaction between LSP1 and myosin II is probably indirect and mediated by F-actin. SiRNA-induced depletion of LSP1 resulted in a ~50% reduction of podosome-localized myosin IIA, but not of cellular myosin IIA levels. LSP1 thus seems to be crucial for the recruitment of myosin IIA to actin structures, possibly based on its actin bundling activity[31]. Immunoprecipitations from lysates of cells expressing LSP1 truncation constructs showed that for efficient binding of actin and myosin IIA, the complete LSP1 C-terminus, comprising both caldmodulin- and villin headpiece-like F-actin binding domains, is required. These results reflect the ability of LSP1 truncation constructs to localize to podosomes. Interestingly, LSP1 was reported to bind Myo1e during Fcγ receptor-driven phagocytosis in macrophages[27], which could suggest a more widespread ability of LSP1 to interact with myosins. This would also be in line with the presence of calmodulin in LSP1 immunoprecipitates, as calmodulin can also function as a light chain for unconventional myosins, including Myo1e[58]. Of note, the LSP1 C-terminal half is more conserved (85% identity) than the N-terminal half (53% identity) between mice and humans[59], pointing to less conserved regulatory functions of the N-terminus.

Interestingly, previous results showed that siRNA-induced depletion of myosin IIA led to enhanced lifetime of podosomes, similar to depletion of supervillin[20]. Reduced recruitment of myosin IIA subsequent to LSP1 depletion would thus also be expected to lead to higher lifetimes of podosomes. However, we now show that depletion of LSP1 leads to reduced lifetime of podosomes. Possible explanations for this could be that (1) LSP1 supports actin bundling, and its depletion thus affects podosome architecture/stability and indirectly also actin turnover due to an increase of available actin filaments; (2) LSP1 depletion also results in enhanced recruitment of supervillin to peripheral podosomes, which could contribute to their enhanced turnover; (3) LSP1 might also play an additional role in podosome formation and/or stabilization through recruitment of an interaction partner such as WASP[54] that is crucial for these processes[8]. Apparently, LSP1 does not perform the same role as supervillin in myosin II activation and podosome turnover. Supervillin is a myosin II hyperactivator, is prominently recruited to dissolving podosomes and leads to enrichment of phospho-myosin light chain at these podosomes. Podosome dissolution is thus likely to depend on supervillin-

induced myosin IIA hyperactivation. In contrast, LSP1 is associated with only moderate myosin II activation (see Fig. 5f), and even the high enrichment of LSP1 at precursors does not trigger podosome dissolution.

The presence of several myosin regulators was tested in cross-immunoprecipitations of LSP1 and myosin IIA. Indeed, L-MLCK, calmodulin and Ser19 phosphorylated myosin light chain (pMLC), but not ROCK1, ROCK2, RhoA, protein phosphatase 1 or 2, were coprecipitated. These results showed that LSP1 interacts with a variety of specific myosin regulators. The membrane-associated protein supervillin forms another hub for actomyosin[23] and also binds regulators of myosin activity, including L-MLCK[20], and calmodulin, as shown here. Of note, pMLC levels in supervillin (SV1-830) precipitates were significantly higher compared to LSP1 precipitates, confirming supervillin's role as a myosin IIA hyperactivating protein[20], and indicating that LSP1 does not share this ability. Importantly, LSP1 and supervillin not only interact with the same subset of myosin regulators but can also compete for them, as expression of SV1-830 led to a redistribution of L-MLCK, calmodulin and pMLC in cells, from the leading edge towards the trailing edge. This competition results in the formation of subcellular zones of different pMLC levels, leading to a symmetry break in actomyosin activity. Of note, actomyosin symmetry breaking can be achieved by differential recruitment of myosin II isoforms A and B[5]. As macrophages express predominantly myosin IIA[7], an alternative way would be differential recruitment of myosin II regulators with different activities, as shown here for LSP1 and supervillin[20].

Interestingly, cell polarization was recently shown to emerge as a result of competition for G-actin between two actin networks, branched filaments at protrusive sites and unbranched actin bundles of the cortex, which also act as a sink for myosin II. Moreover, only intermediary, but not extreme, levels of myosin II were shown to support cell polarization, with insufficient levels of cortical myosin II activity resulting in the formation of multiple protrusions and thus in loss of overall polarization[42]. Considering that LSP1 functions as an actin bundler[31] that supports moderate levels of myosin II activity, the observed formation of multiple protrusive sites upon LSP1 knockdown seems to be in line with this model and could be explained by the respective reduction of cortical myosin contractility.

It is thus noteworthy that (1) LSP1 knockdown leads to more extreme values for circularity and aspect ratio of macrophages, indicating an uncoupling of local protrusion from cell elongation and thus from productive cell polarization, and (2) these aberrancies in cell morphology can be rescued by expression of SV1-830, which shows a more widespread localization, also extending to the cell periphery, upon LSP1 knockdown. It is thus conceivable that the altered distribution of overexpressed supervillin in LSP1 knockdown cells helps to restore the necessary levels of myosin contractility at the cortex that contribute to proper cell polarization.

Investigating the underlying basis for the differential localization of LSP1 and supervillin, we found that actin isoforms show distinct cellular patterning in macrophages, as previously reported for neurons[60], fibroblasts and endothelial cells[39]. Specifically, β- and γ-cytoplasmic actin show a decreasing gradient from the cell periphery or leading edge towards the cell center or trailing edge, consistent with earlier results[33]. In contrast, α-cardiac actin shows an inverse gradient. These distributions were reminiscent of those of LSP1 and supervillin, respectively. Indeed, supervillin mostly colocalized to rearward regions of polarized cells that show a high α-cardiac over β- or γ-actin ratio.

Quantification of LSP1 and supervillin immunoprecipitations showed that LSP1 bound only ~2/3 of the overall amount of actin, compared to supervillin, probably reflecting the fact that

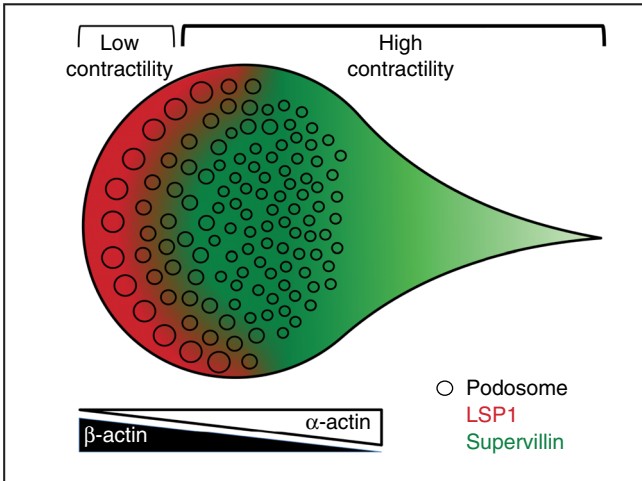

**Fig. 10** Model of actin isoform-based actomyosin symmetry break in macrophages. Actin isoforms localize along inverse gradients in macrophages, with β-actin enriched at the cell periphery and α-actin enriched at the cell interior. Preferential binding of β-actin leads to LSP1 recruitment to structures with a high β- over α-actin ratio, while supervillin is recruited to structures showing a high α- over β-actin ratio. As LSP1 is associated with moderate levels of myosin II activation, whereas supervillin is a myosin II hyperactivator, this leads to a symmetry break in actomyosin-based contractility and to polarization of the cell, establishing zones of low contractility at the front and of high contractility at the back. Large precursor podosomes at the front and smaller successor podosomes at the back show respective enrichments of β-actin and LSP1 versus α-actin and supervillin, which is likely crucial for their different dynamic behaviour

supervillin contains three functional actin-binding sites[23], whereas LSP1 likely contains two. Importantly, LSP1 bound even less of α-cardiac actin, resulting in ~1/3 of α-cardiac actin levels, compared to those coprecipitated with supervillin. This differential binding to actin isoforms by LSP1 was confirmed by cosedimentation assays using pure proteins, with LSP1 preferentially binding to β-actin filaments, and only binding at ~50% reduced levels to α-cardiac actin filaments. Similar experiments performed with the three isolated, GST-fused actin binding regions of supervillin, on the other hand, showed no difference in cosedimentation with β- or α-cardiac actin filaments. Collectively, these results pointed to LSP1, its preferential binding of β-actin, and the presence of subcellular regions containing different ratios of actin isforms, as the drivers of the observed asymmetry in LSP1 and supervillin distribution in cells. Indeed, interfering with the pattern of subcellular actin isoform distribution by depletion of α-cardiac actin resulted in a redistribution of supervillin away from successor podosomes to myosin IIA-positive filaments at the cell cortex. A possible explanation for this phenomenon could be that, in the absence of α−actin-rich successor podosomes, and in the presence of endogenous LSP1, which blocks the access to β-actin-rich precursor podosomes, supervillin is recruited to the cell cortex through its myosin II binding region. Of note, depletion of β-actin did not result in discernibly altered distribution of LSP1, indicating additional mechanisms that stabilize LSP1 at the cell periphery. Still, the altered distribution of supervillin upon LSP1 knockdown clearly shows that preferential binding of β-actin by LSP1 restricts the access of supervillin to this actin pool at the cell periphery. Collectively, these data suggest that the cellular pattern of actin isoforms forms the basis for the differential distribution of both myosin regulatory proteins in cells, thus leading to the establishment of distinct zones of actomyosin activity (Fig. 10).

Different actin isoform ratios are also likely to impact on podosome subpopulations, namely precursors at the leading edge[41] and the more rearward-localized successors[20]. Considering the different biochemical activities of actin isoforms, e.g., differences in polymerization and depolymerization rates, with especially β-actin showing higher dynamics[61,] and also in rheological properties[62], this could, in part, explain the different dynamic behavior of podosomes, with β-actin-rich precursors being more prone to growth and fission, and α-cardiac actin-rich successors being smaller and longer-lived.

In this context, it should be noted that podosome clusters in control cells contain both leading edge-associated, large and highly dynamic precursors and also smaller and more stable successors. In contrast, LSP1 knockdown cells form several clusters, which are often not leading edge-associated, and contain podosomes that show high dynamics and comparatively small size, thus combining characteristics of both precursors and successors. A further contributing factor could be the reduction of LSP1 and myosin IIA at podosome-connecting actin filaments, potentially leading to less coherence within the podosome array and thus giving rise to increased cluster formation. It should thus be worthwhile to investigate these points in detail by using live cell imaging and superresolution microscopy.

Differences between actin isoforms are especially found in their N-termini, the most notable being addition of an aspartic acid residue at the start of the α-cardiac actin sequence[34,35], which is exposed on the surface of the filament[63]. It would thus be interesting to model the F-actin binding regions of supervillin[24] and LSP1 and to determine the structural basis especially for the preferred binding of LSP1 for β-actin. Furthermore, the differential distribution of actin isoforms is likely to impact not only on LSP1 and supervillin, but also on other actin-associated proteins, as hypothesized previously for cofilin[37] and profilin[38].

Another interesting point concerns the establishment of the differential pattern of actin isoforms. One possibility would be localized translation of respective mRNAs. Indeed, the 3′UTR of the β- but not of the γ-actin transcript contains a sequence that binds to zipcode binding protein (ZBP1), which facilitates transcript targeting[34,64]. However, considering that local enrichment of LSP1 and supervillin to the leading and trailing edges is highly dynamic (Supplementary Fig. 8; Supplementary Movie 5), also other mechanisms such as anisotropic self-organization of the cytoskeleton[65], are likely to contribute. Moreover, flow of components such as actin and vinculin has been demonstrated to organize podosomes on the mesoscale[66]. Conceivably, anisotropic flow of actin isoforms during podosome turnover could form a pattern that is used for differential recruitment of LSP1 and supervillin.

Collectively, we show here that LSP1 is a multiple-level regulator of macrophage migration, mechanosensing and invasion, and provide a detailed molecular explanation for the reported effects of LSP1 in immune cell (dys-)regulation. LSP1 competes with supervillin, another actomyosin hub protein, for binding of myosin regulators, and also for actin isoforms, in particular β-actin. Actomyosin symmetry breaking in macrophages is thus achieved by differential recruitment of two actomyosin machineries that support different levels of myosin activity, with the underlying pattern generated by the distribution of actin isoforms.

## Methods
**Isolation and culture and transfection of cells**. Human peripheral blood monocytes were isolated from buffy coats (kindly provided by Frank Bentzien, University Medical Center Eppendorf, Hamburg, Germany) and differentiated into macrophages, as described. Briefly, human heparinized blood is carefully added to Lymphocyte Separation Medium (LSM, PAA Laboratories, Cat. No. J15-004) and

centrifuged for 30 min at 4 °C and 450 × $g$ of speed. Buffy coat containing mononuclear cells and platelets is carefully recovered and monocytes are then isolated using magnetic anti-CD14 beads and separation columns (Miltenyi Biotec, Cat. No. 139-050-201). Approval for the analysis of anonymized blood donations was obtained by the Ethical Committee of the Ärztekammer Hamburg (Germany). Cells were cultured in RPMI-1640 (containing 100 units/ml penicillin, 100 μg/ml streptomycin, 2 mM glutamine and 20% autologous serum) at 37 °C, 5% $CO_2$ and 90% humidity. Monocytes were differentiated in culture for at least seven days, under addition of 20% human autologous serum. For transfection experiments, differentiated macrophages, at days 10–14 of culture, were transiently transfected using the Neon® Transfection System (Thermo Fisher Scientific, Waltham, MA), an electroporation-based system, with standard settings (1000 V, 40 ms, 2 pulses) and a concentration of (100 nM) for LSP1 siRNA, (50 nM) for α-cardiac actin siRNA, (50 nM) for β-actin siRNA and 0.5 μg per 1 × $10^5$ cells for plasmids.

**Expression constructs and siRNA.** The human cDNA of LSP1 (Accession No. for NCBI sequence BC001785) was purchased from Open Biosystem (Catalog Number MHS4771-99611141). LSP1 full length and subdomains were subcloned into pEGFP-C1 empty vector (Clontech) by using the following primers: forward 1 (5-AAATTTAAACTCGAGCCATGGCGGAGGCTTCG AG-3′) and reverse 1 (5′-AAATTTAAAGAATTCCTAGGGGAGCGGGCCCCC-3′) for the full length; forward 1 and reverse 2 (5-AAATTTAAAGAATTCCTAGACCA AGGGGCTGGGTG-3′) for the N-terminus; forward 2 (5′-AAATTTAAACTC-GAGC CTTGGAG GGGACCATCGAACAG-3) and reverse 1 for the C-terminus; forward 2 and reverse 3 (5-AAATTTAAA-GAATTCCTAGGTCTTGGAGCCTCCCTTC-3) for the caldesmon 1/2; forward 3 (5′-AAATTTAAACTCGAGCCTCATCAACAATTAAGAG CACCC-3) and reverse 1 for the villin-headpiece 1/2, containing XhoI (#FD0694, Fermentas) and EcoRI (#FD0274, Fermentas) sites for forward and reverse primers, respectively. The C-terminus was also subcloned into pmCherry-C1 empty vector using the above mentioned restriction sites. The GFP-supervillin 1-174, 1-830 and supervillin-RFP constructs were a kind gift of E. Luna (University of Massachusetts, USA)[67]. pLifeact-TagGFP2 and pLifeact-TagRFP were purchased from Ibidi (Martinsried, Germany). siRNAs specific for human LSP1 were purchased from Eurofins Genomics (#1, 5′-UGGAGACAUGAGCAAGAAA-3′)[68] and from Dharmacon (#2, ON-TARGETplus individual siRNA J-012640-05), with siRNA targeting firefly luciferase mRNA (D-001210-02-20, Dharmacon) used as negative control. Knockdowns were achieved 96 h post transfection. siRNA specific for α-cardiac actin was purchased from Dharmacon (ON-TARGETplus individual siRNA J-012015-05) and knockdown was achieved after 72 h. siRNA specific for β-actin was purchased from Ambion (Silencer Select Pre-designed individual siRNA ID # s230680) and knockdown was achieved after 72 h.

For protein expression in E.coli, LSP1 full length and C-terminal subdomain were subcloned into pEXP5-CT/TOPO TA vector (V960-06, Invitrogen, Thermo Fisher Scientific) with a C-terminal fused 6xHis-tag using the following primers: forward 1 (5′- ATGGCGGAGGCTTCGAGTGACCC-3′) and reverse 1 (5′-GGGAGCCGGGCCCCCT TCCACAA-3′) for the full length; forward 2 (5′-ATGTTGGAGGGGACCA TCGAACAGAG-3′) and reverse 1 for the C-terminus. GST-fused supervillin fragments (171-342; 343-571 and 570-830) were a kind gift of E.Luna (University of Massachusetts, USA)[67]. GST empty vector pGEX-2T was purchased from GE-Healthcare (Buckinghamshire, UK) and used as control for in vitro assay.

**Proteins for in vitro assays.** GST and 6xHis fusion proteins/fragments were expressed after induction in E.coli (BL21) cells, purified with gluthathione-Sepharose$^{TM}$ (GE-Healthcare, Buckinghamshire, UK) or Ni-NTA agarose beads (Qiagen, Hilden, Germany) and dialyzed overnight at 4 °C respectively against the following dialysis buffer (100 mM KCl, 2 mM $MgCl_2$, 1 mM DTT, 40 mM PIPES pH 7.0 and 3 mM $NaN_3$) or (10 mM imidazole pH 7.0, 75 mM KCl, 0.2 mM DTT, 0.2 mM EGTA, 0.01% NP-40 and 3 mM $NaN_3$). Following dialysis, purified proteins/fragments were snap-frozen in liquid nitrogen and stored at −80 °C. Human platelet non-muscle actin (85% β/15% γ, 99% pure, # APLH99) and rabbit skeletal muscle myosin motor protein (full length, 90% pure, # MY02) were purchased from Cytoskeleton, Inc. (Denver, USA), whereas bovine α-cardiac actin 99% pure (# 8201-01) was purchased from Hypermol (Bielefeld, Germany).

**Antibodies and staining reagents.** The following antibodies were used for immunofluorescence (1:100 dilution for primary and 1:200 for secondary, unless otherwise stated), immunoprecipitation, Western blots (1:2000 dilution for primary and 1:5000 for secondary, unless otherwise stated) or PLA assays, as indicated: rabbit polyclonal- (HPA019693), goat polyclonal- (PAB18566) or mouse monoclonal (610734) LSP1 antibodies were purchased, respectively, from Atlas Antibodies (Stockholm, Sweden), Abnova (Taipei, Taiwan) and BD Transduction Laboratories; mouse monoclonal anti-pan actin antibody (MAB1501, 1:5000 for WB) was purchased from Merck Millipore; mouse IgG1 monoclonal anti-human cytoplasmic β-actin (β-CYA)(MCA5775GA) and mouse IgG2b monoclonal anti-human cytoplasmic γ-actin (γ-CYA) (MCA5776GA) antibodies from AbD Serotec (Biorad) and diluted 1:5000 for WB; rabbit polyclonal anti-α cardiac actin (PA5-21396, 1:5000 for WB and 1:50 for IF) and rabbit polyclonal anti-α smooth muscle

actin antibodies (ABT1487) from Invitrogen (ThermoFisher Scientific) and Merck Millipore respectively; rabbit polyclonal (M8064) and mouse monoclonal anti-myosin IIA (60233-1-Ig) antibodies from Sigma-Aldrich (St Louis, MO) and Proteintech, respectively and diluted 1:5000 for WB; rabbit polyclonal phospho-specific (pS20) anti-myosin light chain antibody (ab2480) from Abcam; rabbit polyclonal anti-myosin light chain kinase (MLCK) (ST1657) antibody from Calbiochem (1:50 dilution for IF, 1:1000 for WB); mouse monoclonal anti-calmodulin (# 05–173) antibody from Merck Millipore; rabbit polyclonal anti-ROCK1 (#4035) and mouse monoclonal anti-ROCK2 (610623) antibodies from Cell Signaling and BD Transduction Laboratories, respectively; mouse monoclonal anti-human PP1α catalytic subunit (MAB6105) and polyclonal rabbit anti-PP2α catalytic subunit (#2038) antibodies from R&D Systems and Cell Signaling respectively and diluted 1:1000 for WB; rabbit monoclonal anti-RhoA (ab187027) antibody from Abcam; mouse monoclonal anti-vinculin (V9264) and mouse monoclonal anti-β tubulin (T4026) antibodies from Sigma-Aldrich; rabbit polyclonal anti-GAPDH (10494-1-AP, 1:5000 for WB) antibody from Proteintech; monoclonal mouse anti-GFP antibody (632380, 1:5000 for WB) from Clontech; normal mouse (sc-2025) and rabbit (sc-2027) IgG were purchased from Santa Cruz and used as negative control for PLA assay and immunoprecipitation. HRP-conjugated AffiniPure donkey anti-goat IgG (H + L) (705-035-147) from Jackson ImmunoResearch Laboratories, whereas HRP-linked F(ab')$_2$ fragment donkey-anti rabbit (NA9340v) and HRP-linked sheep-anti mouse IgG (H + L) (NA931v) from GE Healthcare. Alexa Fluor-488 AffiniPure goat anti-rabbit IgG (H + L) (111-545-003) from Jackson ImmunoResearch Laboratories whereas Alexa Fluor-568 goat anti-rabbit IgG (H + L) (A-11011), Alexa Fluor-647 goat anti-rabbit IgG (H + L) (A-21244), Alexa Fluor-568 goat anti-mouse IgG1 (A-21124), Alexa Fluor-647 goat anti-mouse IgG1 (A-21240) and Alexa Fluor-568 goat anti-mouse IgG2b (A-21144) from Invitrogen (Thermo Fisher Scientific). F-actin was stained using Alexa-Fluor-405 (ab176752, Abcam), Alexa Fluor-488 (A12379, Molecular Probes) or Alexa Fluor-568 (A12380, Molecular Probes) phalloidin, as indicated. Coverslips were mounted in Mowiol (Calbiochem) containing DABCO (25 mg/ml; Sigma-Aldrich) as anti-fading reagent and sealed with nail polish. The following antibodies and immunofluorescence reagents were purchased from Abberior and used specifically for STED imaging: Abberior STAR 580-conjugated goat anti-mouse IgG (2-0002-005-1), Abberior STAR RED-conjugated goat anti-rabbit IgG (2-0012-011-9), Abberior STAR 635P-conjugated phalloidin (2-0205-007-0) and Abberior Mount Liquid Antifade (4-0100-005-0).

**Protein sample preparation and immunoblotting.** To evaluate the association of LSP1 with the cortical cytoskeleton, protein lysates were progressively extracted by increasing concentrations of Triton X-100. Cells were washed twice in prewarmed PBS and scraped from dishes in buffer (150 mM NaCl, 1% Triton X-100, 50 mM Tris-HCl (pH 8.0), with protease and phosphatase inhibitors (Roche)). After 5 min on ice with constant agitation, samples were collected in tubes and rotated at 4 °C for 10 min before centrifugation (15 min, 10.000 × $g$, 4 °C). Supernatants were collected, whereas pellets were resuspended in the same volume used in the first step with the addition of 1% Triton X-100 to the initial lysis buffer, for a final concentration of 2% Triton X-100. Samples were pipetted thoroughly and let rotate at 4 °C for 10 min before centrifugation (15 min, 10.000 × $g$, 4 °C). This procedure was sequentially repeated until a final concentration of 5% Triton X-100 was reached, and supernatants were saved. Last pellets were resuspended in RIPA buffer (150 mM NaCl, 1% Triton X-100, 0.5% sodium deoxycholate, 0.1% SDS, 50 mM Tris–HCl (pH 8.0)) and vortexed. Equal volumes of protein samples were then mixed with 4× Laemmli sample loading buffer and examined by standard immunoblotting procedure using NuPAGE 4–12% Bis-Tris gels (Invitrogen, Thermo Fisher Scientific), iBlot2 dry blotting system (Thermo Fisher Scientific) and the above-mentioned primary and HRP-conjugated antibodies, as indicated. When needed, nitrocellulose membranes were mild-stripped by extensive washing with buffer (200 mM glycine, 3.5 mM SDS, 1% tween20, (pH 2.2)) before reblocking and reprobing membranes with primary and secondary HRP-conjugated antibodies. Protein bands were visualized by using Super Signal Pico or Femto kit (Pierce) and X Omat AR films (Kodak). Results were scanned and protein band intensities quantified with Fiji distribution of ImageJ (NIH, Bethesda, MD).

**Actin co-sedimentation assay.** Co-sedimentation binding assays were performed by adding LSP1-6xHis or GST-supervillin fragments to pre-polymerized F-actin according to manufacturer's instructions (Actin binding protein biochem kit$^{TM}$, # BK001, Cytoskeleton, Inc.). Briefly, 20 μM of G-actin was polymerized for 1 h at 24 °C, then mixed to respective purified tagged proteins and left 30 min at 24 °C. The F-actin: protein molar ratio was respectively 16 μM: 4 μM for LSP1-6xHis and 10 μM: 2 μM for GST-supervillin fragments. Samples were then centrifuged at 100,000 × $g$ for 1 h at 24 °C with a Sorvall Discovery M120 ultracentrifuge (Thermo Fisher Scientific) and supernatant and pellet resuspended in SDS loading buffer to reach equal volumes. For direct comparability, equal amounts of sample volumes were analyzed by SDS-PAGE, and bands were visualized by Coomassie blue staining using the Colloidal Blue Staining kit (LC6025, Invitrogen, Thermo Fisher Scientific). Gels were then scanned and protein band intensities quantified with Fiji distribution of ImageJ (NIH, Bethesda, MD).

**Myosin co-sedimentation assay**. For co-sedimentation assay with sedimentable rabbit skeletal muscle myosin, BSA and purified LSP1- fl or C-terminus-6xHis were mixed together with myosin in the following buffer[23] (50 mM KCl, 10 mM MgCl₂, 20 mM HEPES pH 7.1, 0.1% β-mercaptoethanol, 0.01 mM EGTA) to a molar ratio of 1: 7 (myosin: purified protein) and incubated for 90 min at 24 °C. Samples were then centrifuged at $100,000 \times g$ for 90 min at 24 °C and processed, as described above (see actin co-sedimentation assay).

**Immunoprecipitation**. Immunoprecipitation of endogenous protein, as well as of GFP constructs was performed according to manufacturer's instructions (Miltenyi Biotech) with modifications. Cells were washed twice in prewarmed PBS and scraped from dishes in buffer (500 mM NaCl, 2% Triton X-100, 50 mM Tris-HCl (pH 8.0), with protease and phosphatase inhibitors (Roche)). After 20 min on ice with constant agitation, samples were collected in tubes and let rotate at 4 °C for 10 min before centrifugation (30 min, $10,000 \times g$, 4 °C). Supernatants were collected and protein concentration measured with Pierce BCA assay (Thermo Fisher Scientific). Equal amount of protein samples were precleared with a mixture of μMACS Protein A/G Microbeads, rotated for 30 min at 4 °C and loaded into prewashed μMACS columns. Eluates were collected and incubated with a mixture of μMACS Protein A/G Microbeads+specific antibody as indicated or μMACS anti-GFP Microbeads, for 2 h with constant rotation at 4 °C. After incubation, protein samples were loaded into prewashed μMACS columns, washed once with starting lysis buffer, twice with RIPA buffer [150 mM NaCl, 1% Triton X-100, 0.5% sodium deoxycholate, 0.1% SDS, 50 mM Tris-HCl (pH 8.0)], twice with high-salt RIPA buffer (500 mM NaCl, 1% Triton X-100, 0.5% sodium deoxycholate, 0.1% SDS, 50 mM Tris–HCl (pH 8.0)) and once with 20 mM Tris-HCl (pH 7.5) before elution with buffer (50 mM Tris–HCl (pH 6.8), 50 mM DTT, 1% SDS, 0.005% bromphenol blue, 10% glycerol). Samples were then mixed with 4× Laemmli sample loading buffer, heated 10 min at 95 °C and examined by standard immunoblot. Normal mouse and rabbit IgG, matching the isotype species of the tested primary antibody, were used as control for immunoprecipitation of endogenous protein, whereas GFP-IP of pEGFP-C1 empty vector was used as control for immunoprecipitation of GFP overexpressed constructs. To test the role of F-actin in the binding of LSP1 to non-muscle myosin IIA, untreated macrophages were scraped in lysis buffer (2% Triton X-100, 150 mM NaCl, 50 mM Tris-HCl (pH 8.0), 5 mM ATP, 5 mM MgCl₂) in the presence of protease inhibitor cocktail (Roche). The sample was subsequently centrifuged at $10,000 \times g$ for 30 min at 4 °C and the supernatant (input) was split into four different samples, two of which were loaded with latrunculin A to a final concentration of 10 μM. All tubes were then incubated with a mixture of μMACS Protein A/G Microbeads+rabbit polyclonal anti-myosin IIA (M8064, Sigma-Aldrich), rotated at 4 °C for 1 h, loaded into prewashed μMACS columns, washed three times with μMACS lysis buffer (1% Triton X-100, 150 mM NaCl, 50 mM Tris–HCl (pH 8.0)) + 10 mM ATP/MgCl₂ where needed, once with 20 mM Tris–HCl (pH 7.5) + 10 mM ATP/MgCl₂ where needed, and then eluted with buffer (50 mM Tris–HCl (pH 6.8), 50 mM DTT, 1% SDS, 0.005% bromphenol blue, 10% glycerol). Samples were then mixed with 4× Laemmli sample loading buffer, heated 10 min at 95 °C and examined by Coomassie blue staining and standard immunoblot.

**Proximity ligation assay**. Two weeks-old human primary macrophages were seeded on glass coverslips and incubated overnight. Coverslips were then pre-fixed in −20° C cold methanol for 1 sec prior to actual fixation with 3.7% formaldehyde/PBS for 15 min, and permeabilisation in 0.5% Triton X-100 for 5 min. The assay was performed according to manufacturer's instructions (Duolink In Situ, Sigma Aldrich), with an antibody concentration of (0.4 ng/μL).

**Immunostaining**. Macrophages seeded on glass coverslips were pre-fixed in −20 °C cold methanol for 1 sec prior to actual fixation with 3.7% formaldehyde/PBS for 15 min, and permeabilisation in 0.5% Triton X-100 for 5 min. For better visualization of the podosomal F-actin network for STED imaging, cells were partially extracted/fixed using a solution of 3.7% formaldehyde/0.2% Triton X-100 for 5 min prior to actual fixation with 3.7% formaldehyde/PBS for 10 min and permeabilisation in 0.5% Triton X-100 for 5 min. After fixation samples were blocked with antibody diluting solution (0.05% Triton X-100/PBS + 2% BSA + 0.1% Sodium Azide) + 5% NGS. After staining with specific antibodies or labelling reagents, coverslips were mounted in Mowiol (Calbiochem) containing DABCO (25 mg/ml; Sigma-Aldrich) as anti-fading reagent and sealed with nail polish. The staining of actin isoforms was performed according to the protocol previously by Dugina et al.[39], with minor modifications. Briefly, macrophages seeded on glass coverslips were fixed in 1% formaldehyde dissolved in prewarmed RPMI for 30 min and then permeabilised with methanol for 5 min at −20 °C. Primary antibodies against specific actin isoforms were mixed 1:100 and probed in combination, as indicated. Images of actin isoform staining represent the Z-projection of the several confocal planes that comprise podosomes.

**Microscopy**. Images of fixed samples were acquired with a Leica TCS SP5 AOBS confocal laser scanner equipped with an oil-immersion HCX PL APO lambda blue 63× NA 1.4 objective and 3× HyD, 1× PMT detectors. Microscope control and image acquisition were performed with Leica LAS AF software (Leica). Time lapse movies were acquired using the above mentioned Leica TCS SP5 AOBS or with UltraVIEW VoX system (Perkin Elmer) equipped with a Nikon Eclipse Ti

microscope body with perfect focus function, a Yokogawa CSU X1 confocal spinning disk, an oil-immersion 60× Apo TIRF (corr.) NA 1.49 objective and a Hamamatsu EM-CCD C9100-50 camera. Microscope control and image acquisition were performed with Volocity 6.1.1 software (Perkin Elmer). The above-mentioned microscopes were equipped with environmental chambers to allow temperature, humidity and CO₂ control. 3D STED imaging of fixed samples were carried out with Abberior 4 channel easy 3D STED superresolution microscope. This system comprises a Nikon Ti-E microscope body, and an oil-immersion 60× P-Apo NA 1.40 objective. For excitation, 561 and 640 nm pulsed lasers were combined with the 775 nm STED pulsed laser. For detetection, 4 avalanche photodiodes (APD) were used in gated mode. Microscope control and image acquisition were performed with Abberior Imspector software (Abberior Instruments).

**Protrusion force measurements**. Protrusion force measurements were performed as described[50,69]. Briefly, Atomic Force Microscopy (AFM) measurements were performed using silicon nitride cantilevers (MLCT-AUHW, Veeco Instruments) with a nominal spring constant of 0.01 N/m mounted on a NanoWizard III AFM (JPK Instruments) coupled to an inverted optical microscope (Axiovert 200, Carl Zeiss). To prepare Formvar sheets, ethanol-cleaned glass slides were dipped into a Formvar solution of 0.5% (w/v) ethylene dichloride (Electron Microscopy Science) for a few seconds and the solution was emptied from the film casting device using a calibrated flow. A Formvar film was detached from dried slides by contact with water and was left floating at the surface. Acetone-washed 200-mesh nickel grids (EMS) were arranged on the floating film, picked up coated with the film onto another glass slide and then air-dried. To evaluate the thickness of the Formvar sheet, the border of the Formvar that remained on the glass slide after removing the grids was imaged in contact mode by AFM. For AFM experiments of living cells seeded on Formvar-coated grids, a temperature-controlled chamber was used (Petri dish heaterTM, JPK Instruments) and the culture medium was supplemented with 10 mM HEPES (pH = 7.4) (Sigma-Aldrich). Images were recorded in contact mode in liquid at scanning forces lower than 1 nN. Cell-induced protrusions were imaged with a pixel resolution of 256 or 512 pixels at line rate around 2 Hz. Forces exerted by single podosomes were derived from the topographical data of podosome-induced protrusions[69] and each cell was attributed the median force value of its podosomes. Briefly, the deformation profile of each protrusion was measured on the AFM image using an ImageJ macro and, combined with the ring radius values, led to the determination of the deformation height h. This was converted to force for each podosome by the relation $F = C_0 \frac{E}{1-\nu^2} \frac{h_f^3}{r_t^2} h$ where the biaxial Young's modulus of Formvar $E/(1-\nu^2)$ is 2.3 GPa[50], $C_0 \approx 2.7$ is a geometric coefficient evaluated from numerical simulations[69] and the film thickness $h_f$ and ring radius $r_t$ were measured for each series of experiments by AFM and immunofluorescence respectively.

**Live cell imaging**. For live cell imaging, cells were transfected with respective constructs, as indicated, seeded on glass bottom live cell dishes (Ibidi) at sub-confluent concentration and incubated overnight before starting live cell imaging acquisition. To evaluate podosome cluster and cell movement together with podosome lifetime in LSP1 depletion conditions, cells were transfected with control siRNA and siRNA against LSP1. After 72 h, cells were re-transfected with pLifeact-TagGFP2 or pLifeact-TagRFP and seeded on glass bottom live cell dishes (Ibidi) at subconfluent concentration. After overnight incubation cells were imaged for at least 1 h. 24 cells from four different donors were evaluated.

**Macrophage 3D invasion and LSP1 localization**. 3D collagen I invasion of LSP1-depleted macrophages was performed and analysed as described. Briefly, primary human macrophages were transfected with specific siRNA, copolymerised with rat tail collagen (2.5 mg/ml; Becton Dickinson) and embedded in rat tail collagen I (2 mg/ml) containing 10 ng/ml Macrophage-Colony Stimulating Factor (M-CSF, Relia Tech, Wolfenbüttel, Germany). Four days after transfection, brightfield micrographs of invaded areas were acquired (4 images per well) and the number of cells that invaded the surrounding matrix was counted using Fiji cell counter. Four wells per donor and three donors in total were analysed ($n = 16$). For LSP1 3D localization, macrophages were cotransfected with GFP-LSP1 and pLifeact-TagRFP, embedded in rat tail collagen I (2 mg/ml) and incubated overnight before confocal imaging.

**Image analysis**. Images were post-processed and analysed using Volocity 6.1.1 for Mac (Perkin Elmer) and/or Fiji distribution of ImageJ version 1.51. For color-coded analysis (Fig. 2), each single frame from a time-lapse video was color-coded using the "Temporal color code" tool of Fiji and according to the "Rainbow RGB" LUT. Podosome clusters/cells tracks of LSP1-depleted cells were generated by plotting the XY coordinates of the center of mass of podosome cluster or cells, as calculated by Volocity. The coordinates were imported into GraphPad, and the XY values pruned to have all trajectories starting at X=0, Y=0. Podosome lifetime of LSP1-depleted cells was evaluated by calculating the difference between the time of appearance (fission event or de novo formation) and disappearance (fusion event or dissolution) of ten podosomes per cell (24 cells from four different donors per condition). Cell shape descriptors such as "aspect ratio" (AR), "circularity" (C) and "cell area" were measured using Fiji. Specifically, aspect ratio is calculated as (major axis×minor

axis⁻¹) therefore representing solely the degree of elongation, whereas circularity is calculated as [4π*(area × perimeter⁻²)], thus representing the degree of similarity to a circumference with a value ranging from 0 to 1 (perfect circle). To analyze podosome oscillations upon GFP-LSP1 overexpression or knockdown (Fig. 3), the intensities of five individual podosomes per cell (6 cells from two different donors per condition) were measured at every timepoint (20 sec) of a 10 min-long video, using Fiji; intensity values were normalized by subtracting the respective avarage intensity. The number and height of peaks were measured using the "Area under the curve" tool from GraphPad by setting the normalized average intensity as baseline. A single peak was defined as the highest Y-value between two consecutive Y-values below the baseline and the peak height was measured as difference between the highest Y-value of a peak and respective baseline; peak within the shoulder of another peak were not taken in consideration. F-actin core and myosin IIA intensities in LSP1 knockdown experiments (Fig. 4) were measured based on an ImageJ macro[70]. In addition, myosin IIA ROIs were generated by applying the "dilate" tool of Fiji to the previously detected F-actin core ROIs. For proximity ligation assay analysis, images of fixed samples were acquired with confocal Leica TCS SP5 microscope and PLA spots counted using the "FindMaxima" tool of Fiji. 30 μm-long intensity profiles of proteins in Figs. 6,7,9 were measured starting from cell cortex toward the inside of 10 cells from three different donors. In order to allow direct comparability among different protein profiles, the values were interpolated using the "cubic spline" tool from GraphPad and setting the same number of values for all curves; values were then corrected for respective average intensity and normalized to F-actin (or β-actin) intensity values when needed. Ratiometric analysis of actin isoforms was performed with Fiji: confocal Z-planes comprising podosomes were projected onto a single focal plane with averaged intensities, with contrast normalized and enhanced to include max 0.3% of saturated pixels in the podosome area; the "α-cardiac isoform channel was then divided by "β-" or"γ-actin isoform" channel, respectively, and the output calculated as 32-bit floating point values. Ratiometric values of generated images were thresholded from 0 to 2 after applying the "mpI-inferno" LUT. For better visualization, the ratiometric images were "smoothened" by running a 1 pixel Median filter.

**Statistical analysis and softwares used**. All data were processed with Microsoft Excel 2011 and GraphPad Prism 5 for Mac OSX. Data are presented as mean ± s.e. m. if not otherwise stated in the respective legends. Statistical comparisons are performed using one-sample t-test, two-tailed unpaired t-test or one-way ANOVA with Bonferroni's multiple comparison test as appropriate. Detailed descriptions and values can be found in respective legends and Suppl. Data 1. Statistically significant differences are indicated by single/multiple asterisks (****$P < 0.0001$; ***$P < 0.001$; **$P < 0.01$; and *$P < 0.05$). Correlation analysis was calculated and plotted using a correlation plot and linear regression line. Pearson correlation coefficients (r) are shown in the plots.

**Data availability**. The authors declare that the data supporting the findings of this study are available within the paper and its supplementary information files, and are available from the authors upon reasonable request.

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

## Acknowledgements

We thank Frank Bentzien (UKE transfusion medicine) for buffy coats, Elizabeth Luna for the supervillin-RFP, GFP-SV1-830 and GST supervillin constructs, Andrea Mordhorst for excellent technical assistance, Antonio Virgilio Failla and Bernd Zobiak (UKE microscope facility) for excellent help with imaging, Christian Wurm (Abberior Instruments) for help with 3D STED, Christophe Thibault and Christophe Vieu for AFM microscopy, Karim El azzouzi for critical reading of the manuscript, and Isabelle Maridonneau-Parini and Martin Aepfelbacher and his group for continuous support. This work is part of the doctoral thesis of P.C. This work has been supported by grants from Fondation pour la Recherche Médicale and ANR14-CE11-0020-02 to I.M.-P., and by Deutsche Forschungsgemeinschaft (LI925/2-2), the Wilhelm Sander Stiftung (2014.135.1), and the European Union's Seventh Framework Programme (FP7/2007–2013) under grant agreement no. FP7-237946 (T3Net) to S.L.

## Author contributions

P.C. designed and performed experiments and co-wrote the manuscript, C.W. performed 3D invasion experiments, A.B. and R.P. performed protrusion force microscopy, S.L. designed experiments and wrote the manuscript.

## Additional information

**Competing interests:** The authors declare no competing financial interests.

