## [Peer Review File · Nature Communications]

Reviewers' comments:

Reviewer #1 (Remarks to the Author):

The major novel claims in this manuscript are that LSP1 is a hub protein that interacts with myosin IIA and myosin IIA-associated regulatory proteins and that the LSP1 hub competes with supervillin, another hub protein in the actomyosin cortex to regulate podosome function in macrophages. A second intriguing and important claim is that "LSP1 and supervillin compete for actin isoforms, especially alpha-cardiac actin, and that the cellular pattern of actin isoforms forms the basis for the differential distribution of both myosin hub proteins in cells." If fully substantiated, these data will be very important conceptually for the podosome field and, by extension, to others interested in actomyosin-based mechanosensing.

The data presented in the paper are, for the most part, visually stunning and the product of apparently well-performed and carefully controlled experiments. The authors use state-of-the-art fluorescence microscopy techniques, including atomic force microscopy, confocal light microscopy and high-resolution STED, to show that LSP1 localizes to the anterior cap structure of podosomes, mechanosensing actin-based organelles on the ventral surfaces of macrophages, and to actin filament-based cables linking these structures. All experiments are carefully quantified and analyzed for statistical significance. In agreement with published results, the authors report that the LSP1 C-terminus is primarily responsible for the association of LSP1 with podosomes. They extend the previous work by showing that the last 33 amino acids of LSP1, which contain two villin-like headpiece structures are sufficient for targeting. LSP1 function is explored after ~50% reduction of the protein using each of two siRNAs that target different LSP1 sequences. LSP1 depletion is shown to enhance podosome dynamics, to increase the speed and distance covered of macrophages undergoing 2D migration and 3D invasion, and to reduce protrusion force at the ventral surface. LSP1 localizes primarily to precursor podosomes that form close to the cell periphery, where its signal overlaps with that of myosin IIA, although the overlap observed by STED microscopy is not overwhelming. LSP1 knockdown reduces myosin IIA recruitment to these structures. Localizations and co-immunoprecipitations of overexpressed fluorescent LSP1 and the N-terminus of supervillin show differences in associated proteins and confirm the previous report from this laboratory that supervillin is primarily associated with so-called successor podosomes at and near the rear of the podosome field in migrating macrophages. Alpha-cardiac actin also is shown to localize preferentially at successor podosomes. These results support the previous report of biochemical differences between precursor and successor podosomes and establish LSP1 as a marker for the precursors.

MAJOR COMMENTS

1. The current data in this manuscript support, but do not prove, the most important conclusions drawn. To prove that "LSP1 interacts with myosin IIA and recruits it to podosomes", or to say that "LSP1 acts as a hub for myosin IIA", or especially to conclude that "efficient binding" of actin and myosin IIA requires the entire LSP1 C-terminus, one needs to see some data for direct binding. The manuscript currently lacks any in vitro binding experiments; the supporting data for the claimed interactions are co-immunoprecipitations (coIPs) for most of the novel interactions and a proximity ligation assay (PLA) for LSP1 and myosin IIA.

The problem with relying on PLA and coIP data is that each give positive results with indirect interactions. The authors say that the PLA signals indicate proximity of ≤ 40 nm, which may be a reasonable average for this technique. However, the Duolink kit from Sigma-Aldrich used for these experiments relies on DNA-tagged secondary antibodies. Each IgG is approximately 15 nm from Fab to Fc. With two primary and two secondary antibodies in the experiment, the antigens could

be as far apart as 60 nm (4 x 15 nm) and still give a positive signal. A distance of 40 nm to 60 nm is the length of 10 to 15 actin monomers (the size of a short actin filament) or a myosin II head (S1, 16.5 nm) bound to a different actin monomer within the same short filament. LSP1 and supervillin each bind directly to F-actin, so a reasonable alternative explanation for the authors' results is that both the PLA and coIP assays are indicating the presence of separate populations of actin filaments without necessarily indicating a causal role for LSP1 or supervillin in establishing the basis for the different populations. In the case of supervillin, direct binding has been demonstrated for both F-actin and the myosin IIA heavy chain, so the authors only need to demonstrate a direct interaction between LSP1 and myosin IIA to call LSP1 a "hub" for actin and myosin II and to carry on with (most of) their current claims. Constructs and a protocol for the preparation of bacterially expressed LSP1 proteins (full-length, N-terminal and C-terminal) have been published (Jongstra-Bilen et al., 1992, J Cell Biol 118, 1143-53) in the (currently uncited) paper that demonstrates direct binding of LSP1 to F-actin.

2. Similarly, if the authors want to state that they have shown "preferred binding of supervillin for alpha-cardiac actin", they have to show a direct binding *in vitro*.

3. Also, if the authors want to retain the claim that "establishment of these subcellular zones [LSP1- versus supervillin-based podosomes] is based on the differential distribution of actin isoforms", they need to show that a knockdown of alpha-cardiac actin prevents the dichotomy that they see.

4. The current co-IP experiments suffer from an additional problem. Macrophages are transfected with GFP-tagged LSP1 or GFP-supervillin N-terminus (SV 1-830), solubilized with 500 mM salt, 2% Triton X-100, etc. and then anti-GFP is used to precipitate the associated proteins, which are analyzed solely by immunoblotting. The levels of protein overexpression are unclear in each case. GFP-LSP1 is much smaller in size (~80 kDa) than GFP-SV 1-830), raising the question about whether some of the observed differences in co-sedimenting proteins has to do with relative protein percentages. The difference observed for pMLC, relative to the myosin II heavy chain, looks real, but as noted by the authors, that is likely caused by supervillin-mediated myosin hyper-activation that they have already described. What ARE the levels of each GFP-protein, relative to the level of the corresponding endogenous protein and relative to each other, based on total lysate protein concentrations?

Parenthetically, the presence of calmodulin in the coIP pellets is consistent with the previous demonstration of co-IP of MyoIE with LASP1 and with supervillin. Calmodulin is a light chain for many nontraditional myosins, including IE (Jang et al., 2007, J Proteome Res 6, 3718-28). In the case of supervillin, the IP pellet also contained other proteins associated with Triton-insoluble membrane domains (Nebi et al., 2002, J Biol Chem 277, 43399-409). The authors should consider carrying out MudPit proteomics analyses on each of their anti-GFP coIP pellets.

5. Especially in the absence of careful quantifications of the relative levels of the input proteins, the so-called "competition experiments" of proteins co-IPing with the mCherry-tagged LSP1 C-terminus with and without GFP-SV 1-830 (Figures 6E, 6F) are unconvincing. A real competition experiment performed *in vitro* with purified myosin IIA-binding LSP1 and supervillin sequences would be interesting and informative, but at a minimum, these two panels and the text should be deleted.

OTHER COMMENTS AND CRITICISMS

1. Podosome cluster nomenclature. In Figure 2 and in the text on pages 5 and 6, the authors refer to podosome clusters. Are these clusters of precursor podosomes, successor podosomes, or a mix of both? In their previous paper (Ref. 24), they show that the successor podosomes are on the path to dissolution. That could be consistent with the observation of increased podosome dynamics

upon LSP1 reduction because LSP1 levels are reduced in the successor podosomes, but this point is not made specifically in the Discussion. Clarification of the localization(s) of the analyzed "podosome clusters" would be very helpful.

2. On a similar note, did the authors observe reduced lifetimes for BOTH precursor and successor podosome in LSP1-knockdown macrophages? Based on LSP1 localizations and their hypothesis about supervillin-mediated control of function in the successor podosomes, one might predict that the LSP1 reduction would selectively affect the precursors.

3. In Figure 3, panels I, J and K, the locations of all the cells are marked by surrounding red boxes. At the magnification at which the images are shown, the underlying black dots of cells cannot be seen; only red boxes are visible. To be consistent with the journal's policy about being able to inspect all relevant data, I recommend that the authors submit images WITHOUT the red boxes as part of their already-extensive collection of supplementary data.

4. RNAi technology. 100 nM is a very high concentration for siRNA experiments; 10 to 20 nM are the normal levels. Because the knockdowns are only 50% to 60%, I suspect that there is a problem with either the quality of the siRNAs or with transfection of macrophages. For future studies, the authors might try obtaining siRNAs with protective groups on the ends, in case RNA degradation is the reason that they have to use such high siRNA concentrations. Alternatively, maybe this is a typo, with 10 nM intended?

5. Immunostaining. The macrophages are said to have been pre-fixed in -20°C methanol for 1 sec prior to actual fixation with 3.7% formaldehyde/PBS. One second is a VERY short period of time. Is this correct? If so, how was the step accomplished and why did it have to be performed? Significant treatment times with cold methanol preclude phalloidin staining of actin filaments, so this may be a key step for obtaining the results reported herein.

6. Legend for Suppl Figure 1, line 18. The legend cites "Yellow lines (I,II) in (A)" but obviously should say "Yellow lines (1,2) in (F)".

Reviewer #2 (Remarks to the Author):

In this manuscript, Cervero et al. identify LSP1 as a novel component of the podosome cap, mainly of the dynamic precursors at the leading edge. The authors show that LSP1 is involved in podosome cluster dynamics, podosome oscillations, and macrophage 2D and 3D migration. Furthermore, the potential interaction with myosin and myosin interactors is studied as well as potential competition with supervillin for myosin and its interactors. Finally, a differential distribution for actin isoforms has been uncovered that potentially drives the differential localization of supervillin, LSP1 and the myosin interactors.

The manuscript is interesting and relevant for the field of podosomes and macrophage migration as well as the field of actomyosin cytoskeleton regulation, is very well-written and include a good range of complementary methodologies ranging from super resolution STED to co-IP and PLA, to high quality fluorescence microscopy. However, the paper in my view should be improved on a number of aspects before it can be considered for publication.

The quantifications are a bit rudimentary (e.g. profiles through cells), lack statistics (e.g. profile through single cell), vary between experiments (podosome lifetime in Fig 2K, podosome oscillations Fig 4) or simply do not help with interpreting the results (e.g. regression analysis of mean displacement plots). More robust quantifications (see below) are needed to strengthen the stated conclusions.

Finally, the conclusions seem to contradict previous results from the same group. More specifically, whereas the authors have previously shown that myosin activation primarily occurs at supervillin successor podosomes (see REF#24, Bhuwania et al J Cell Sci 2012), they here claim that LSP1 recruits these interactors to the leading edge and only by overexpressing the N terminal domain of

supervillin, a relocation to the rear is occurring. It is to me unclear whether LSP1 and Supervillin recruit myosin or the other way around. Taken this manuscript and the Bhuwania paper together one can make an argument for both possibilities. Ideally, experiments should be conducted to clarify this issue. At the very least, if the authors have an explanation for this, a thorough discussion should be included. On the same vein, the localization of MLCK and pMLC under normal conditions is surprising considering ref#24, where they show that pMLC localized to supervillin positive podosomes under steady state conditions and that MLCK did not show any particular preference for precursors or successors. The data presented here seem to directly contradict those previous findings, and the authors should at least provide an explanation for these apparent contradictions.

Major comments:

1) The authors state that LSP1 is mainly located at peripheral podosomes. While this may be true for polarized macrophages (although not all images show a clear gradient), the specific enrichment of LSP1 at peripheral podosomes in live resting macrophages (Suppl Fig. 2A-L and also Fig1 A-B) is not convincing, since also the actin signal is higher in the peripheral podosomes in most of the images. The authors should perform a more rigorous quantification of the localization of LSP1 instead of the presented profile plots. I suggest to take the LSP1/Actin intensity ratio and correlate this to the distance from the cell edge/front.

2) It is unclear how the linear regression of the mean displacement plot helps to quantify the cluster dynamics data from Figure 2. It is also unclear how the numbers were derived and what these numbers mean (6.1, 6.6, 3 μ m, page 6, line 148). Mean squared displacement is usually only applied when one wants to discriminate between random, confined or directed motion. This could be interesting but the interpretation and analysis of these results remains superficial. Also, the authors make a distinction between mobility and velocity (page 6, line 151), but the authors give insufficient explanation on the difference between these terms (is there a difference?) and interpretation of the associated results.

3) The authors show that knockdown of LSP1 results in less recruitment of myosin IIA to podosomes and to a lower lifetime of podosomes. Yet, they have published before that myosin IIA knockdown results in longer lifetime. The authors should discuss this apparent discrepancy. Also, did knockdown of LSP1 result in a general decreased recruitment of myosin IIA or only at the peripheral podosomes where LSP1 can be found?

4) The 3D migration experiments are insufficiently analyzed, which makes it difficult to compare them to the 2D data. For example, was speed also increased in 3D after LSP1 knockdown? Could the authors observe enhanced podosome dynamics in 3D after LSP1 knockdown? Did the authors also observe a precursor/successor distribution in 3D? These are all questions that are relevant to put the 3D results into context. I realize 3D migration experiments are challenging but if they cannot be fully interpreted, they are maybe better as supplementary observations.

5) It is difficult to link the overexpression experiment to the rest of the results. Why was oscillation period quantified here and not lifetime, while for the knockdown lifetime was quantified and not oscillation period. I would recommend the authors to quantify similar parameters in different experiments to be able to contrast and compare results. Also, protrusion force was only measured for the knockdown and not for the overexpression. It would also be interesting to analyze protrusion force differences for precursors/successors under the given conditions.

6) On page 7, line 187, the authors state that the reduced protrusion force is in line with their findings on podosome oscillations. It is unclear how this relationship is established. Are short period oscillations indicative of less force? On what basis do the authors state this?

7) The authors conclude that LSP1 interacts with myosin IIA based on the co-precipitation and the PLA assay, which is a preliminary conclusion. These results suggest that they are likely together in

a complex but direct binding is not demonstrated here.

This applies even more to the competition assays. Both the co-IP (Figure 6E) results and supervillin overexpression experiments (Figure 7) can have other explanations aside from competition. In vitro binding assays should give better answers if direct interactions are to be investigated.

8) The relocalization of MLCK (Figure 7C,G) and pMLC (Figure 7S,W) is not convincing based on the images shown and the single intensity profiles. The authors should provide a better quantification (e.g. ratio of component/actin with respect to cell edge, as suggested also above, or intensity values in region of interests (a few pixels large) progressively going from the cell edge towards the cell center) to convincingly show that relocalization is occurring.

9) The gradient localization of gamma actin in Fig. 8a is not convincing. I rather see a higher signal above and below the chosen cross-section line. A second cross-section line chosen at a different location through the cluster is likely to give a different profile. A better quantification should be provided. Also, it is unclear how the r values from the regression analysis in panel E of Fig 8a should be interpreted. Based on their subcellular localization it is surprising that beta and gamma actin also show an increased co-IP with supervillin. The authors should discuss this more thoroughly.

Minor comments

- The conclusion that the dynamics of actin and LSP1 are similar is preliminary. The frame rate that is used is pretty low and only one podosome is shown in suppl fig 2.
- Page 8, line 230. The authors conclude that endogenous LSP1 is being coprecipitated with the C terminal LSP1 construct. Could that band just be the C-terminal construct being recognized by the antibody?
- A model of how myosin IIA is recruited to these different hubs would be very helpful.
- Mechanosensing in the title seems an over-interpretation of the results since no direct mechanosensing is shown in the paper.

We are very grateful for the reviewers' constructive comments and have tried to address the raised points as closely as possible. Please find a point-by-point response below. The respective changes are marked in red in the manuscript.

In response to minor point 4 of referee 2, and to also reflect the important role of LSP1 in symmetry breaking, as outlined in the response to point 2 of referee 1, we have now altered the title to "Lymphocyte specific protein-1 regulates mechanosensory oscillation of podosomes and actin isoform-based actomyosin symmetry breaking".

Reviewer #1

Major comments:

1. The current data in this manuscript support, but do not prove, the most important conclusions drawn. To prove that "LSP1 interacts with myosin IIA and recruits it to podosomes", or to say that "LSP1 acts as a hub for myosin IIA", or especially to conclude that "efficient binding" of actin and myosin IIA requires the entire LSP1 C-terminus, one needs to see some data for direct binding. The manuscript currently lacks any in vitro binding experiments; the supporting data for the claimed interactions are co-immunoprecipitations (coIPs) for most of the novel interactions and a proximity ligation assay (PLA) for LSP1 and myosin IIA.

The problem with relying on PLA and coIP data is that each give positive results with indirect interactions. The authors say that the PLA signals indicate proximity of ≤ 40 nm, which may be a reasonable average for this technique. However, the Duolink kit from Sigma-Aldrich used for these experiments relies on DNA-tagged secondary antibodies. Each IgG is approximately 15 nm from Fab to Fc. With two primary and two secondary antibodies in the experiment, the antigens could be as far apart as 60 nm (4×15 nm) and still give a positive signal. A distance of 40 nm to 60 nm is the length of 10 to 15 actin monomers (the size of a short actin filament) or a myosin II head (S1, 16.5 nm) bound to a different actin monomer within the same short filament. LSP1 and supervillin each bind directly to F-actin, so a reasonable alternative explanation for the authors' results is that both the PLA and coIP assays are indicating the presence of separate populations of actin filaments without necessarily indicating a causal role for LSP1 or supervillin in establishing the basis for the different populations. In the case of supervillin, direct binding has been demonstrated for both F-actin and the myosin IIA heavy chain, so the authors only need to demonstrate a direct interaction between LSP1 and myosin IIA to call LSP1 a "hub" for actin and myosin II and to carry on with (most of) their current claims. Constructs and a protocol for the preparation of bacterially expressed LSP1 proteins (full-length, N-terminal and C-terminal) have been published (Jongstra-Bilen et al., 1992, J Cell Biol 118, 1143-53) in the (currently uncited) paper that demonstrates direct binding of LSP1 to F-actin.

Thank you for these very good points and suggestions. We have performed in vitro binding experiments using bacterially expressed His-tagged LSP1 full length, N-terminal and C-terminal constructs and commercially available rabbit skeletal muscle myosin. Following the protocol by (Chen et al., J Biol Chem, 2003), myosin filaments were precipitated by ultracentrifugation, and co-precipitation of LSP1 constructs was assessed. We found coprecipitation of a small amount of C-terminal LSP1. However, this was not significant and might be due to the presence of residual actin in a $\approx 90\%$ pure myosin preparation. Moreover, immunoprecipitation of LSP1 in the presence of latrunculin, leading to a decrease of

coprecipitating actin (filaments) also showed a reduction in LSP1 (new Suppl. Fig. 7B-D). The influence of LSP1 on myosin recruitment is thus probably indirect, although clearly pronounced (see influence of LSP1 knockdown on myosin recruitment to podosomes; Fig. 4). Therefore, it is now mentioned in the paper that the interaction between LSP1 and myosin IIA is probably indirect. Accordingly, we have removed the term “myosin IIA hub protein” from the text and also from the title. LSP1 is now mostly referred to as a myosin II regulator. The paper by (Jongstra-Bilen et al., 1992) is now also cited on p.13. See also referee 2, point 7.

2. Similarly, if the authors want to state that they have shown "preferred binding of supervillin for alpha-cardiac actin", they have to show a direct binding in vitro.

Previous coprecipitations showed that the amount of alpha-actin coprecipitated with supervillin 1-830 is higher than the amount coprecipitated with LSP1. We interpreted this as a preferred binding of supervillin for alpha-actin. However, an alternative explanation could be that LSP1 has a preferred binding for beta-actin. Performing the requested in vitro experiments, we found that this latter explanation is indeed the correct one. We performed F-actin cosedimentation assays using pure beta-actin and alpha-cardiac actin, as well as bacterially expressed and purified His-tagged LSP1. We observed a highly significant difference (N=5; $P < 0.0001$) between cosedimentation of LSP1 with beta-actin filaments (84%) and with alpha-cardiac actin filaments (45%). Similar experiments performed with the three isolated, GST-fused actin binding regions of supervillin (aa 171-342; 343-571; 570-830) on the other hand, showed no difference in cosedimentation with beta- or alpha-cardiac actin filaments.

The respective results are now presented as the new Fig. 8 and new Suppl. Fig. 8, mentioned in the Results (pp. 10,11) and Discussion sections (p.15), while the F-actin cosedimentation assay is now described in the Materials and Methods section. The term “preferred binding of supervillin for alpha-cardiac actin” has been changed to “preferred binding of LSP1 to beta-actin”. Importantly, the preferred binding of LSP1 for beta-actin does not change the scenario for actin-isoform-based symmetry breaking by LSP1 and supervillin, but rather highlights the critical role that LSP1 is playing in this process.

3. Also, if the authors want to retain the claim that "establishment of these subcellular zones [LSP1- versus supervillin-based podosomes] is based on the differential distribution of actin isoforms", they need to show that a knockdown of alpha-cardiac actin prevents the dichotomy that they see.

We have now performed siRNA-based knockdown of alpha-cardiac actin in primary human macrophages and analysed the subcellular distribution of supervillin. Moreover, as the experiments performed in response to point 2 showed that LSP1 preferentially binds beta-actin, thus potentially blocking access of supervillin to beta-actin-rich structures, we have also analysed supervillin distribution in macrophages depleted for LSP1.

Knockdown of alpha-cardiac actin resulted in a redistribution of SV1-830 away from successor podosomes to linear, myosin IIA-positive filaments at the cell cortex. A possible explanation for this phenomenon is that, in the absence of successor podosomes with a high alpha over beta actin ratio, and in the presence of endogenous LSP1, which localizes preferentially to beta-actin rich precursor podosomes, and thus blocking access of supervillin to these structures, supervillin is now recruited to the cell cortex through its myosin II binding ability. Indeed, the distribution of SV1-830 in alpha-cardiac actin depleted cells strongly resembles that of the isolated myosin IIA binding domain (SV1-174).

We reasoned that if the observed dichotomy of LSP1 and supervillin distribution is based on the preferred binding of LSP1 to beta actin (with supervillin showing no preference and binding equally well to alpha and beta; see point 2), depletion of LSP1 should result in a redistribution of supervillin towards the cell periphery. We thus performed siRNA-based knockdown of LSP1 using two individual sequences and analysed supervillin distribution in these cells. We observed a clear redistribution of SV1-830 that now extended also to the beta-actin-rich precursor podosomes at the leading edge/cell periphery. This was further corroborated by fluorescence intensity measurements based on the analysis of 10 cells. Furthermore, we often observed a loss of cell polarization typically associated with expression of SV1-830 (Bhuwania et al., 2012), with cells now showing only small or absent trailing edges and a more radially symmetric distribution of the SV1-830 signal.

Collectively, these data show that the observed dichotomy is based on the differential distribution of actin isoforms and the preferred binding of LSP1 to beta actin. Interference with this pattern by either depletion of alpha-cardiac actin or LSP1 results in clearly altered distribution of supervillin and also in reduced cell polarization. These data are now included as the new Fig. 8 and are mentioned in the Results (p.11) and Discussion sections (p.15).

4. The current co-IP experiments suffer from an additional problem. Macrophages are transfected with GFP-tagged LSP1 or GFP-supervillin N-terminus (SV 1-830), solubilized with 500 mM salt, 2% Triton X-100, etc. and then anti-GFP is used to precipitate the associated proteins, which are analyzed solely by immunoblotting. The levels of protein overexpression are unclear in each case. GFP-LSP1 is much smaller in size (~80 kDa) than GFP-SV 1-830), raising the question about whether some of the observed differences in co-sedimenting proteins has to do with relative protein percentages. The difference observed for pMLC, relative to the myosin II heavy chain, looks real, but as noted by the authors, that is likely caused by supervillin-mediated myosin hyper-activation that they have already described. What ARE the levels of each GFP-protein, relative to the level of the corresponding endogenous protein and relative to each other, based on total lysate protein concentrations?

We would like to point out that all bands in the immunoprecipitation experiments were already normalized to the respective GFP protein bands. Still, we agree with the referee that determination of the overexpression ratios is indeed informative. We have, therefore, performed Western blots of macrophage lysates and detected GFP (LSP1 GFP, supervillin 1-830 GFP, endogenous LSP1 and endogenous supervillin). We find that LSP1-GFP levels in lysates are a factor of 1.1 higher than the endogenous protein levels, and that GFP-SV1-830 levels are a factor of 2.1 higher than endogenous supervillin levels. Considering the transfection rate of primary human macrophages for the respective GFP fusion proteins of 17% (LSP1) and 19% (SV1-830), LSP1-GFP is 6.5 times overexpressed, while SV 1-830 is 11 times overexpressed. Similarly, comparing the GFP-fused forms of LSP1 and SV, the LSP1-GFP/SV 1-830-GFP ratio was determined as 1.6. This information is now included in the manuscript text (p.9).

Parenthetically, the presence of calmodulin in the coIP pellets is consistent with the previous demonstration of co-IP of MyoIE with LASP1 and with supervillin. Calmodulin is a light chain for many nontraditional myosins, including IE (Jang et al., 2007, J Proteome Res 6, 3718-28). In the case of supervillin, the IP pellet also contained other proteins associated with Triton-insoluble membrane domains (Nebl et al., 2002, J Biol Chem 277, 43399-409). The authors should consider carrying out MudPit proteomics analyses on each of their anti-GFP coIP pellets.

Thank you for pointing this out. The role of calmodulin as a light chain for unconventional myosins is now also mentioned in the paper, along with the respective citation (p.13). We are indeed considering proteomic analyses of the anti-GFP coIPs. This should be interesting, but we believe it is outside of the scope of the current manuscript.

5. Especially in the absence of careful quantifications of the relative levels of the input proteins, the so-called "competition experiments" of proteins co-IPing with the mCherry-tagged LSP1 C-terminus with and without GFP-SV 1-830 (Figures 6E, 6F) are unconvincing. A real competition experiment performed in vitro with purified myosin IIA-binding LSP1 and supervillin sequences would be interesting and informative, but at a minimum, these two panels and the text should be deleted.

We agree with the referee. Based on the data gathered in response to point 1, an indirect binding of LSP1 and myosin IIA is more likely. We have therefore removed the panels of former Figure 6E and 6F, and also the respective text from the manuscript.

Other comments:

1. Podosome cluster nomenclature. In Figure 2 and in the text on pages 5 and 6, the authors refer to podosome clusters. Are these clusters of precursor podosomes, successor podosomes, or a mix of both? In their previous paper (Ref. 24), they show that the successor podosomes are on the path to dissolution. That could be consistent with the observation of increased podosome dynamics upon LSP1 reduction because LSP1 levels are reduced in the successor podosomes, but this point is not made specifically in the Discussion. Clarification of the localization(s) of the analyzed "podosome clusters" would be very helpful.

This is a good point, and we agree with the referee on the correlation between podosome cluster dynamics and podosome subgroups. We would like to point out, though, that successor podosomes are actually long-lived and only the most rearward ones dissolve upon acquiring higher levels of myosin IIA, which probably drives a feed-forward cycle of supervillin-dependent myosin IIA activation, thus leading to successor dissolution.

The clusters in the control cells mostly comprise a single field of podosomes, as podosomes cover most of the ventral surface in quiescent, radially symmetric cells or are recruited to the leading edge in migrating macrophages. Clusters in control cells thus contain both precursors and successors. In contrast, LSP1 knockdown cells form several clusters, which are not necessarily leading edge-associated, show higher dynamics and less variation in size. Still, successor podosomes are more stable than precursors, so the highly dynamic podosomes in LSP1 knockdown cells do not fit the criteria for either precursors or successors. A further contributing factor could be the reduction of LSP1 and myosin IIA at connecting cables, leading to less coherence within the podosome array and thus giving rise to increased cluster formation. This is now also mentioned in the Discussion (p.16).

2. On a similar note, did the authors observe reduced lifetimes for BOTH precursor and successor podosome in LSP1-knockdown macrophages? Based on LSP1 localizations and their hypothesis about supervillin-mediated control of function in the successor podosomes, one might predict that the LSP1 reduction would selectively affect the precursors.

As outlined in the response to point 1), podosomes in LSP1 knockdown cells are more uniform, both in terms of size and life time and do not fit the criteria for either precursor or successor podosomes. LSP1 depletion thus seems to affect both precursors and successors. As

a consequence, we could only measure the life times for this single existing type of podosomes.

3. In Figure 3, panels I, J and K, the locations of all the cells are marked by surrounding red boxes. At the magnification at which the images are shown, the underlying black dots of cells cannot be seen; only red boxes are visible. To be consistent with the journal's policy about being able to inspect all relevant data, I recommend that the authors submit images WITHOUT the red boxes as part of their already-extensive collection of supplementary data.

Thank you for this good suggestion. The panels from former Fig. 3I,J,K are now shown without red boxes as the new Suppl. Fig. 5 D'-F'.

4. RNAi technology. 100 nM is a very high concentration for siRNA experiments; 10 to 20 nM are the normal levels. Because the knockdowns are only 50% to 60%, I suspect that there is a problem with either the quality of the siRNAs or with transfection of macrophages. For future studies, the authors might try obtaining siRNAs with protective groups on the ends, in case RNA degradation is the reason that they have to use such high siRNA concentrations. Alternatively, maybe this is a typo, with 10 nM intended?

Many thanks for this good suggestion. In our experience, siRNA-induced knockdown in primary human macrophages requires 50-100 nM of siRNA. This could be due to the high cytoplasm content/volume of these cells, compared to other cell types. It has been routinely performed in the lab with a variety of different target proteins, with knockdown efficiencies ranging from 50%-98%. It thus most likely reflects a real requirement for these high concentrations in this cell type, and not a quality issue of the respective siRNAs.

5. Immunostaining. The macrophages are said to have been pre-fixed in -20°C methanol for 1 sec prior to actual fixation with 3.7% formaldehyde/PBS. One second is a VERY short period of time. Is this correct? If so, how was the step accomplished and why did it have to be performed? Significant treatment times with cold methanol preclude phalloidin staining of actin filaments, so this may be a key step for obtaining the results reported herein.

1 sec of prefixation with methanol is indeed correct. In our experience, it helps to get a clearer immunofluorescence signal for cytoskeleton-associated proteins, especially in primary macrophages, which are rich in cytoplasm. For this, coverslips with cells are dipped for 1 sec into pre-cooled methanol in glass dishes kept in a -20°C freezer, subsequent to removing excess medium with a tissue. As mentioned by the referee, prolonged treatment with methanol (> 20 sec) precludes phalloidin staining of F-actin, so we strictly stick to the 1 sec prefixation.

6. Legend for Suppl Figure 1, line 18. The legend cites "Yellow lines (I,II) in (A)" but obviously should say "Yellow lines (1,2) in (F)".

Thank you for spotting this. The legend has now been corrected.

Reviewer #2

The quantifications are a bit rudimentary (e.g. profiles through cells), lack statistics (e.g. profile through single cell), vary between experiments (podosome lifetime in Fig 2K, podosome oscillations Fig 4) or simply do not help with interpreting the results (e.g.

regression analysis of mean displacement plots). More robust quantifications (see below) are needed to strengthen the stated conclusions.

Finally, the conclusions seem to contradict previous results from the same group. More specifically, whereas the authors have previously shown that myosin activation primarily occurs at supervillin successor podosomes (see REF#24, Bhuwania et al J Cell Sci 2012), they here claim that LSP1 recruits these interactors to the leading edge and only by overexpressing the N terminal domain of supervillin, a relocation to the rear is occurring. It is to me unclear whether LSP1 and Supervillin recruit myosin or the other way around. Taken this manuscript and the Bhuwania paper together one can make an argument for both possibilities. Ideally, experiments should be conducted to clarify this issue. At the very least, if the authors have an explanation for this, a thorough discussion should be included. On the same vein, the localization of MLCK and pMLC under normal conditions is surprising considering ref#24, where they show that pMLC localized to supervillin positive podosomes under steady state conditions and that MLCK did not show any particular preference for precursors or successors. The data presented here seem to directly contradict those previous findings, and the authors should at least provide an explanation for these apparent contradictions.

We are now including experiments that show that LSP1 probably does not directly bind myosin IIA, as outlined in response to the specific points below. This is in contrast to the direct binding of myosin II by supervillin. Still, the presence of LSP1 is necessary for proper levels of myosin IIA at podosomes and especially at the cell periphery. Earlier, we characterized supervillin as a myosin hyperactivator, as it binds to activated myosin (with binding abrogated in the presence of blebbistatin), and leads to further activation of myosin on top of that basal level. Supervillin thus needs (active) myosin II for its localization, and, once localized, also recruits further myosin molecules (shown in Bhuwania et al., 2012, Fig. 6 and Suppl. Movie 4), while LSP1 is necessary for basal recruitment of myosin.

The localization of MLCK and pMLC in this study is now based on thorough evaluation and quantification of 10 cells from 3 donors, as requested by the referee in point 8 (see below). These localizations are not contradictory to data published in (Bhuwania et al., 2012), as pMLC is shown in this previous publication to localize to all podosomes, although being enriched at supervillin-positive ones (Fig. 7 in Bhuwania et al., 2012). We did not report a localization for endogenous MLCK in (Bhuwania et al., 2012). Please note that the label “MLCK” in Fig. 8D-F of that paper refers to use of a MLCK-specific siRNA, while the actual stainings/signals shown are for F-actin and GFP-supervillin, and that the localization shown in Suppl. Fig. 10 is for a GFP-MLCK construct, not for endogenous MLCK. Even so, also the GFP-MLCK construct shows an enrichment at the leading edge of a polarized cell, compatible with the more robust quantification shown in the current manuscript.

Major comments:

1) The authors state that LSP1 is mainly located at peripheral podosomes. While this may be true for polarized macrophages (although not all images show a clear gradient), the specific enrichment of LSP1 at peripheral podosomes in live resting macrophages (Suppl Fig. 2A-L and also Fig1 A-B) is not convincing, since also the actin signal is higher in the peripheral podosomes in most of the images. The authors should perform a more rigorous quantification of the localization of LSP1 instead of the presented profile plots. I suggest to take the LSP1/Actin intensity ratio and correlate this to the distance from the cell edge/front.

Thank you for this good suggestion. We have now measured the LSP1/F-actin ratio, in both resting and migratory macrophages, along a distance of 30 μm starting at the cell periphery. Fluorescence intensity graphs of 6-10 cells were acquired for each condition. Respective ratio graphs show that LSP1 is indeed enriched over F-actin by a factor of 2-2.5 at the cell edges, with a gradual decline of the ratio towards the inner parts of cells. The zone of LSP1 over F-actin enrichment extends about 5-10 μm from the cell periphery, before the ratio drops below 1 in the more distant cell part. This zone of enrichment is wide enough to contain the belt of precursor podosomes in both resting and migratory cells. These results are now presented in the new Suppl. Figure 1K-M and mentioned in the Results section (p.5).

Furthermore, the discovery of LSP1 showing preferred binding to beta-actin (see referee 1, point 2), combined with the high beta-over-alpha ratio of the peripheral precursor podosomes, is well in line with a preferential enrichment of LSP1 at these structures. The respective results are now presented as the new Fig. 8, mentioned in the Results (pp10,11) and Discussion sections (P.15), while the new F-actin cosedimentation assay is now described in the Materials and Methods section (p.15). The term “preferred binding of supervillin for alpha-cardiac actin” has been changed to “preferred binding of LSP1 to beta-actin”. See also referee 1, point 2.

2) It is unclear how the linear regression of the mean displacement plot helps to quantify the cluster dynamics data from Figure 2. It is also unclear how the numbers were derived and what these numbers mean (6.1, 6.6, 3 μm , page 6, line 148). Mean squared displacement is usually only applied when one wants to discriminate between random, confined or directed motion. This could be interesting but the interpretation and analysis of these results remains superficial. Also, the authors make a distinction between mobility and velocity (page 6, line 151), but the authors give insufficient explanation on the difference between these terms (is there a difference?) and interpretation of the associated results.

We agree that mean square displacement is usually applied in the analysis of random vs. directional migration. We offered the regression analysis as a means to illustrate the fact that the migration of LSP1 knockdown cells deviated to a higher degree from the mean, compared to control cells. The term “mobility” was also used to illustrate this higher deviation from the mean, as a higher velocity of clusters would not necessarily lead to this more random pattern of migration. Still, we agree with the referee that this is not strictly necessary and might even confuse readers. The respective panels G-I were thus removed from Figure 2, as well as their description in text and legends, which includes the term “mobility”.

3.1 The authors show that knockdown of LSP1 results in less recruitment of myosin IIA to podosomes and to a lower lifetime of podosomes. Yet, they have published before that myosin IIA knockdown results in longer lifetime. The authors should discuss this apparent discrepancy.

This is indeed an interesting point. Our previous results showed that siRNA-induced depletion of myosin IIA led to enhanced lifetime of both podosome subpopulations, similar to depletion of supervillin (Bhuwania et al., 2012). Reduced recruitment of myosin IIA subsequent to LSP1 depletion would thus also be expected to lead to higher lifetimes of podosomes. However, as depletion of LSP1 leads to shorter lifetimes, LSP1 has to play an additional role in podosome formation and/or stabilization that is independent of myosin IIA recruitment. This could be achieved through recruitment of an interaction partner such as WASP (Prasad et al., 2012) that has a role in these processes (Linder et al., 1999).

Apparently, LSP1 does not perform the same role as supervillin in myosin II activation and podosome turnover. Supervillin is a myosin II hyperactivator, is prominently recruited to

dissolving podosomes and leads to enrichment of phospho-myosin light chain at these podosomes. Podosome dissolution is thus likely to depend on supervillin-induced myosin IIA hyperactivation. In contrast, LSP1 is only a moderate myosin II activator (see Fig. 6D), and even the high enrichment of LSP1 at precursors does not trigger podosome dissolution. These points are now also mentioned in the Discussion section (p.13,14).

3.2) Also, did knockdown of LSP1 result in a general decreased recruitment of myosin IIA or only at the peripheral podosomes where LSP1 can be found?

LSP1 is typically enriched at precursor podosomes, but is present at all podosomes. Knockdown of LSP1 results in a general decrease of myosin IIA recruitment.

4) The 3D migration experiments are insufficiently analyzed, which makes it difficult to compare them to the 2D data. For example, was speed also increased in 3D after LSP1 knockdown? Could the authors observe enhanced podosome dynamics in 3D after LSP1 knockdown? Did the authors also observe a precursor/successor distribution in 3D? These are all questions that are relevant to put the 3D results into context. I realize 3D migration experiments are challenging but if they cannot be fully interpreted, they are maybe better as supplementary observations.

We agree that the 3D experiments share no direct correlation with the much more elaborate 2D data. They were included to show that LSP1 also has a role in 3D migration. 3D podosomes are difficult to identify (only a combination of markers is effective) and thus to track in living cells. In fact, there is no publication so far describing live cell imaging of 3D podosomes. The architecture of 3D podosomes is also unclear, as well as the question whether they show subpopulations. Clearly, these questions merit a much more detailed examination in future work. We have thus moved the 3D data to the supplementary material (new Suppl. Fig. 5), as suggested by the referee.

5) It is difficult to link the overexpression experiment to the rest of the results. Why was oscillation period quantified here and not lifetime, while for the knockdown lifetime was quantified and not oscillation period. I would recommend the authors to quantify similar parameters in different experiments to be able to contrast and compare results. Also, protrusion force was only measured for the knockdown and not for the overexpression. It would also be interesting to analyze protrusion force differences for precursors/successors under the given conditions.

We agree that results from different experiments should be comparable. We have therefore performed extensive experiments to enhance the comparability of the data. Please note that some of the proposed experiments were not possible to perform for experimental or inherent reasons, as outlined below.

We have performed new podosome oscillation experiments using cells treated with two individual siRNAs or a control siRNA. Analyzing these data, we realized that the differences between the fluorescence intensity curves can be best described by the parameters “number of local peaks”, “height of local peaks” and “frequency distribution of peak height”. For better comparability, we also reanalyzed the data from EGFP controls and LSP1-GFP overexpressing cells. The former Fig.4, now Fig.3, has been restructured accordingly. It now clearly shows that overexpression of LSP1-GFP leads to a 50% enhancement of the number of local peaks, and a slight shift to lower values in the frequency distribution of peak height. SiRNA-mediated depletion of LSP1 moderately reduces the number of local peaks, but leads to a strong increase in peak height and also in a broader distribution of the frequencies of peak

high, i.e. a higher deviation from controls. In sum, LSP1-GFP overexpression leads to enhanced “frequency” of podosome oscillations, while LSP1 depletion results in a more irregular “amplitude”. The data are now shown in the restructured Figure 3 and are mentioned in the Results (p.7) and Discussion (p.12) sections.

We agree that protrusion force measurements between precursor and successor in LSP1 knockdown cells would be very interesting. However, as also described in point 2 of referee 1, all podosomes in LSP1 depleted cells show similar characteristics and a clear distinction between precursors (larger size, fission) and precursors (smaller size, no fission) is not possible. We also tried to measure protrusion force in LSP1-GFP overexpressing cells. However, despite multiple (n=8) attempts, this was unsuccessful for a combination of reasons. For protein knockdown, adherent cells can be treated with a lipofection-based method of siRNA transfection. This is in contrast to overexpression, where cells have to be detached prior to electroporation. The latter method generally results in a strongly reduced transfection efficiency (2-15%). While this would be sufficient for AFM analysis, electroporated cells failed to adhere to the Formvar sheets and died after 24h. We thus tried to culture transfected cells for an intermediate period of 1d on glass before transferring them to Formvar sheets, but the few cells that survived did not form podosomes on Formvar.

6) On page 7, line 187, the authors state that the reduced protrusion force is in line with their findings on podosome oscillations. It is unclear how this relationship is established. Are short period oscillations indicative of less force? On what basis do the authors state this?

We agree that one should be cautious in establishing a correlation between podosome oscillation period and protrusive force. The phrase “consistent with the data on podosome oscillations” has therefore been removed. We are now only stating that the reduction of protrusion force probably indicates “a positive influence of LSP1 on the generation of podosome protrusive force”.

7) The authors conclude that LSP1 interacts with myosin IIA based on the co-precipitation and the PLA assay, which is a preliminary conclusion. These results suggest that they are likely together in a complex but direct binding is not demonstrated here. This applies even more to the competition assays. Both the co-IP (Figure 6E) results and supervillin overexpression experiments (Figure 7) can have other explanations aside from competition. In vitro binding assays should give better answers if direct interactions are to be investigated.

We agree with the referee on these points. We have performed in vitro binding experiments using bacterially expressed His-tagged LSP1 fl, N-terminal and C-terminal constructs and rabbit skeletal muscle myosin IIA. Following the protocol by (Chen et al., J Biol Cell, 2003), myosin filaments formed in vitro were precipitated by ultracentrifugation, and co-precipitation of LSP1 constructs was assessed. We found coprecipitation of a small amount of C-terminal LSP1. However, this was not significant and might be due to the presence of residual actin in a $\approx 90\%$ pure myosin preparation. Moreover, immunoprecipitation of myosin IIA in the presence of Mg^{2+}/ATP and latrunculin, leading to a decrease of coprecipitating actin (filaments), also showed a reduction in coprecipitating LSP1 (new Suppl. Fig. 7B-D). The influence of LSP1 on myosin recruitment is thus probably indirect, although clearly pronounced (see influence of LSP1 knockdown on myosin recruitment to podosomes; Fig. 4). We have therefore amended our statement about an interaction between LSP1 and myosin IIA and removed the term “myosin IIA hub protein” from the paper, and also from the title. (See also referee 1, point 1).

We also agree with the referee in regard to the competition assays. Based on the data gathered also in response to point 1 of referee 1, an indirect binding of LSP1 and myosin IIA is more likely. We have therefore removed the panels of former Figure 6E and 6F, and also the respective text from the manuscript.

8) The relocation of MLCK (Figure 7C,G) and pMLC (Figure 7S,W) is not convincing based on the images shown and the single intensity profiles. The authors should provide a better quantification (e.g. ratio of component/actin with respect to cell edge, as suggested also above, or intensity values in region of interests (a few pixels large) progressively going from the cell edge towards the cell center) to convincingly show that relocation is occurring.

Thank you for this good suggestion. We have now quantified respective intensities of LSP1 and supervillin 1-830, as well as MLCK, calmodulin and phospho-MLC along lines that are a few pixels wide and 30 μm long, starting from the cell edges towards the cell center. Respective mean intensities were calculated for 10 cells from 3 donors. These results indeed show that MLCK and calmodulin intensities are lowered at the cell front and increased towards the rear upon overexpression of SV1-830. This is also accompanied by an increase of pMLC levels at the rear. Respective graphs are now shown in the new panels of Fig. 6D,H,L,P,T,X.

9) The gradient localization of gamma actin in Fig. 8a is not convincing. I rather see a higher signal above and below the chosen cross-section line. A second cross-section line chosen at a different location through the cluster is likely to give a different profile. A better quantification should be provided. Also, it is unclear how the r values from the regression analysis in panel E of Fig 8a should be interpreted. Based on their subcellular localization it is surprising that beta and gamma actin also show an increased co-IP with supervillin. The authors should discuss this more thoroughly.

We agree that the localization of gamma-actin in Fig. 8C does not entirely reflect the fluorescence intensity analysis in Fig. 8D. Still, the cross-section line was not chosen randomly, but along the front-rear axis of the polarized cell. For a more quantitative evaluation, we thus analysed respective fluorescence intensities from 10 different cells, as shown in Fig. 8D. These data clearly show that gamma-actin shows a decrease along the front-rear axis.

The increased binding also of beta- and gamma-actin by supervillin is probably based on a higher overall capacity of supervillin for actin binding, reflecting the fact that supervillin contains three functional actin-binding sites (Chen et al., 2003), whereas LSP1 contains probably only two. This seems also to be supported by the amount of coprecipitated actin, as pan-actin levels coprecipitated by LSP1 correspond to $\sim 2/3$ of those precipitated by supervillin. As the co-IPs were, by their nature, performed from lysates, localization of actin isoforms is not the decisive factor here. The preferred binding of beta-actin by LSP1 is thus even more remarkable and points to a specific affinity for this actin isoform. These points are now discussed in more detail on p.15

Minor comments

1) The conclusion that the dynamics of actin and LSP1 are similar is preliminary. The frame rate that is used is pretty low and only one podosome is shown in suppl fig 2.

Please note that the numbers given in the gallery in Suppl. Fig. 2 indicate the time points of the respective frames, not the frame rate. The frame rate used was 5sec/frame, as indicated in the Materials and Methods section. This frame rate was also used by the Geiger group in (Luxenburg et al., J Cell Sci, 2012) to visualize clear differences in the recruitment of individual podosome components. We are thus very sure that we would have detected differential recruitment of F-actin and LSP1, if this were the case. In contrast, we observed on numerous occasions similar recruitment dynamics for both components at podosomes. The podosome shown in Suppl. Fig. 2 is typical and just one of numerous (n>20) podosomes that were analysed for this.

2) Page 8, line 230. The authors conclude that endogenous LSP1 is being coprecipitated with the C terminal LSP1 construct. Could that band just be the C-terminal construct being recognized by the antibody?

The referee is absolutely correct. The anti-LSP1 antibody recognizes the LSP1 C-terminus. It is thus likely that the band represents the C-terminal construct. The respective sentence has now been removed from the text, and a respective statement has been added to the figure legend.

3) A model of how myosin IIA is recruited to these different hubs would be very helpful.

We agree that a model would be helpful for readers. We have now included a model of myosin recruitment to LSP1 and supervillin-enriched zones, based on the distribution of actin isoforms, and leading to a symmetry break of actomyosin in macrophages. This has been added as the new Fig. 9.

4) Mechanosensing in the title seems an over-interpretation of the results since no direct mechanosensing is shown in the paper.

We understand the referee's reluctance in using the term mechanosensing. Still, oscillatory movement of podosomes and protrusion force-dependent indentation of substratum, both of which are demonstrated here to be regulated by LSP1 (Fig. 3), are intrinsic and accepted parts of podosomal mechanosensing (Labernadie et al. "Protrusion force microscopy reveals oscillatory force generation and mechanosensing activity of human macrophage podosomes", Nat Commun 2014; van den Dries et al., Nat Commun, 2013). In response to the referee's concern, we have now altered the term "mechanosensing" to "mechanosensory oscillation of podosomes", which adequately describes our findings.

Reviewers' comments:

Reviewer #1 (Remarks to the Author):

This manuscript greatly extends the initial report of LSP1 at podosomes (Ref. 67) by characterizing the effects of LSP1 overexpression and knockdown on podosome dynamics and function. The authors go beyond this already publishable achievement to tackle the mechanism by which macrophages transition between a quiescent, circular stage and an elongated, highly motile state. Their overarching hypothesis is that "LSP1 recruits myosin IIA and its regulators... and competes with supervillin... for myosin regulators and also for actin isoforms, notably beta-actin." The revised manuscript further states that "these results show that the cellular pattern of actin isoforms builds the basis for the differential distribution of two actomyosin machineries with distinct properties" and that "these findings form a new concept for cellular symmetry breaking and polarized immune cell migration."

These conclusions are novel and important for our understanding of podosome function and cell migration. LSP1 levels and mutational status have been implicated in immune cell function, HIV infection and breast cancer metastasis, making these studies of interest to clinicians in a number of fields, as well as to basic scientists.

In response to the initial review, the authors have carried out a number of additional experiments and have extensively revised their text. They originally favored the hypothesis that preferential binding of supervillin to cardiac alpha-actin at interior podosomes was responsible for the observed symmetry breaking. A series of binding experiments prompted by the first set of reviews changed the favored mechanism to preferential binding of LSP1 to beta-actin at the cell periphery, which then excluded supervillin from the associated peripheral podosomes. In my opinion, the new experiments have greatly strengthened the manuscript, but the change in mechanistic focus has left cracks in the logic and presentation that can be further improved.

MAJOR COMMENTS

1. Statistical Analyses. The Materials & Methods mentions only Student's two-tailed t tests. This is fine for the many experiments with only two datasets, but one-way ANOVAs need to be performed for the experiments with 3 or more datasets. Eyeball estimates suggest that the existence of statistical significance is not likely to change after re-analysis of Figures 2G, 2H, 2M, 2N, 4E, 4F, 4G, 4H and 8X and Supplemental Figures S4A, S5G and S5H using ANOVA, but the P values might.

2. LSP1 and Circularity. The first mention of the effect of LSP1 on circularity is in the presentation of Figure 8, in which LSP1 knockdown increases the circularity of macrophages that overexpress SV 1-830 GFP. At least, that was my reading of the text and the legend. The legend for Fig. 8X does not specify that the cells being scored were all SV 1-830 overexpressors, but that is the focus of this figure. The legend needs to be clarified.

Assuming that this interpretation is correct, the reader wants to know what effect LSP1 knockdown has on the circularity of macrophages that do NOT overexpress SV 1-830. On the one hand, the cell with LSP1 siRNA in Figure 4 looks rounder than the cells with control siRNA in the same panel. On the other hand, LSP1 siRNAs increase cell speed (Fig. 2M) and cell invasion (Supplemental Figure S5). Both of these properties are generally associated with cell elongation. So, does LSP1 knockdown have opposite effects in untransfected macrophages versus macrophages expressing SV 1-830 GFP? If so, how does that fit with the hypothesis of dueling actomyosin contractile mechanisms? Data on the effect of LSP1 knockdown on circularity should be added to either Figure 2 or Figure 4, and any implications for the mechanism included in the Discussion.

3. Figure 5 and LSP1 Co-immunoprecipitation with non-muscle myosin IIA (now Suppl. Fig. 7C and 7D). The demonstration that LSP1 binding to non-muscle myosin IIA requires the presence of co-bound F-actin is an important point that is currently buried in the supplementary figures. Figure 5 has a lot of white space. I strongly recommend that panels C and D in Supplementary Figure 7 be incorporated into Figure 5 in the main text, probably as Fig. 5E and 5F.

4. Text on pages 8 and 9 (lines 240 - 248), Figure 5 and Supplementary Figure 7. As currently presented the reader is asked to believe that the strongest evidence for a lack of direct binding of LSP1 to macrophage myosin IIA is the absence of co-sedimentation of LSP1 with rabbit skeletal muscle myosin II. In fact, their strongest evidence is the data with non-muscle myosin IIA (currently Suppl. Fig. 7C, discussed above). As the authors discuss for alpha-, beta- and gamma-actin, many proteins exhibit binding preferences for specific protein isoforms, and the myosin II heavy chains differ more than the actin isoforms. For instance, supervillin has been reported to bind only to non-muscle and smooth muscle myosin II proteins and not to rabbit skeletal muscle myosin II (Ref. 28), so the absence of binding of LSP1 to skeletal muscle in Suppl. Fig. 7B is not necessarily relevant to myosin II regulation in macrophages. I would recommend that the authors flip the presentation of the Suppl. Fig. 7C data (after moving it to Fig. 5) and the Suppl. Fig. 7B data in the text, making the less compelling data secondary to the more compelling evidence.

5. Figure 6. The title for Figure 6 is that "LSP1 and supervillin compete for myosin regulators in cells." That is a significant and important question. However, the data show only the change in localization of the interactors after over-expression of either GFP or SV 1-830 GFP. In a paper about LSP1, I would have expected Figure 6 to show shifts in the distributions of MLK, calmodulin and pMLC after over-expression of LSP1 or the LSP1 C-terminus, with the data in the current figure relegated to a supplementary figure. If there are two competing actomyosin-assembling machineries, I would expect to see complementary shifts after over-expression (or under-expression) of EITHER LSP1 OR supervillin. Assuming that to be true, including that data would greatly support the overarching hypothesis of this paper.

6. Figure 8. Similarly, the title for Figure 8 is "Differential subcellular recruitment of LSP1 and supervillin is based on the preferential binding of LSP1 to beta-actin." This is another important question. Yet, the data show only the knockdown of cardiac ALPHA-actin and the results of expressing myosin IIA-binding sequences from supervillin. This data made sense when the prevailing hypothesis was preferential binding of supervillin to cardiac alpha-actin, but since the authors have shot down that hypothesis, this figure becomes an interruption of the logical flow, given the new biochemical results. The current data are supportive, but, as with Figure 6, the reader is primed to learn what happens after a knockdown of beta-actin with or without overexpression of LSP1 or the LSP1 C-terminus. For instance, does LSP1 or cardiac alpha-actin re-localize?

7. Figure 8A. A second reason for wanting to know about the results of a beta-actin knockdown is that one could argue that the knockdown of cardiac alpha-actin has had an effect due to a decrease in the total level of cellular actin, not necessarily because of a shift in the ratios of the actin isoforms. Opposite shifts in localization of relevant proteins after beta-actin knockdown, assuming that this doesn't also negatively affect alpha-actin levels, would greatly support the authors' thesis. This brings up the question of whether knockdown of one of the actin isoforms affects the levels of the other cellular actins. Figure 8A shows only a single immunoblot of alpha-cardiac actin and beta-actin after treatment with control or alpha-actin siRNA. The GAPDH loading control may be equivalent, but its overloading makes this assessment difficult to confirm. The numbers below GAPDH presumably refer to the relative levels of alpha-cardiac actin. Gamma-actin is not shown, and there are no quantifications for beta-actin. Levels of all actin isoforms should be quantified for both gamma- and beta-actin knockdowns.

RELATIVELY MINOR POINTS

1. As the authors have learned, co-immunoprecipitated proteins cannot be said to "bind" to each

other; the preferred phrase is "interact with" or "co-associate". Inappropriate "bind" or "binding" claims remain on lines 261 and 433. The biggest overstatement in this regard is the sentence on page 15, lines 427-428, that "these results showed that LSP1 acts as a hub for myosin activity by binding specific myosin regulators." This sentence needs to be deleted or corrected.

2. Text on page 10, lines 305-306 versus Figure 8D. The text says that the SV 1-830 GFP signal in cells with reduced levels of cardiac alpha-actin is redistributed "away from podosomes and to the cell cortex." The figure shows a signal that does appear to be less central than the corresponding signal in Figure 6, in which the cells express endogenous levels of alpha-actin. However, it is still pretty far away from the "cortical" beta-actin and LSP1 signals in control cells. Perhaps "shifted towards the cell periphery" is more accurate than "cortical", unless the authors have better examples for this effect.

3. Supplementary Figure 9. Why is Supplementary Figure 9 and the videos not presented in the Results section? No data should be presented for the first time in the penultimate paragraph of a paper's Discussion. In addition, these data set the stage for idea of cross-talking dynamic actomyosin machineries at the heart of this work.

4. Text, pages 11-12, lines 342 - 345. Conversely, why are conclusions presented in the Results, instead of the Discussion? With the current data, most of these statements also are speculative. For instance, the conclusion that LSP1 and supervillin "compete for actin isoforms, especially beta-actin" is an inference from the actin binding data and SV 1-830 GFP re-organization after knockdown of alpha-actin. Also, the statement that "the cellular pattern of actin isoforms forms the basis for the differential distribution of BOTH (emphasis mine) [LSP1 and supervillin] in cells" is largely unsupported by the current data.

5. Discussion, pages 14 - 15, lines 411 - 423. I'm surprised that the authors don't propose that the primary suggested mechanism for the shorter podosome lifetimes observed after LSP1 depletion is due to the fact that supervillin is no longer excluded from binding to peripheral precursor podosomes, leading to podosome dissolution as described in this laboratory's previous paper on this subject. Given the theme of the paper, I would expect that the loss of LSP1-mediated recruitment of WASP and Arp2/3 would be prefaced by "Alternatively, or in addition".

6. Discussion, page 17, lines 491 - 494. I am confused by the last clause in this sentence, "potentially leading to less coherence within the podosome array and thus giving rise to increased cluster formation." If the array is less coherent, then wouldn't the clusters be more dispersed?

Reviewer #2 (Remarks to the Author):

I appreciate the substantial experimental work and the textual improvements that the authors added to the manuscript.

They adequately answered to my concerns.

I have no further points and do recommend the paper for publication.

Reviewer 1:

This manuscript greatly extends the initial report of LSP1 at podosomes (Ref. 67) by characterizing the effects of LSP1 overexpression and knockdown on podosome dynamics and function. The authors go beyond this already publishable achievement to tackle the mechanism by which macrophages transition between a quiescent, circular stage and an elongated, highly motile state. Their overarching hypothesis is that "LSP1 recruits myosin IIA and its regulators... and competes with supervillin... for myosin regulators and also for actin isoforms, notably beta-actin." The revised manuscript further states that "these results show that the cellular pattern of actin isoforms builds the basis for the differential distribution of two actomyosin machineries with distinct properties" and that "these findings form a new concept for cellular symmetry breaking and polarized immune cell migration."

These conclusions are novel and important for our understanding of podosome function and cell migration. LSP1 levels and mutational status have been implicated in immune cell function, HIV infection and breast cancer metastasis, making these studies of interest to clinicians in a number of fields, as well as to basic scientists.

In response to the initial review, the authors have carried out a number of additional experiments and have extensively revised their text. They originally favored the hypothesis that preferential binding of supervillin to cardiac alpha-actin at interior podosomes was responsible for the observed symmetry breaking. A series of binding experiments prompted by the first set of reviews changed the favored mechanism to preferential binding of LSP1 to beta-actin at the cell periphery, which then excluded supervillin from the associated peripheral podosomes. In my opinion, the new experiments have greatly strengthened the manuscript, but the change in mechanistic focus has left cracks in the logic and presentation that can be further improved.

Thank you for these insightful and constructive comments.

Major comments:

1. Statistical Analyses. The Materials & Methods mentions only Student's two-tailed t tests. This is fine for the many experiments with only two datasets, but one-way ANOVAs need to be performed for the experiments with 3 or more datasets. Eyeball estimates suggest that the existence of statistical significance is not likely to change after re-analysis of Figures 2G, 2H, 2M, 2N, 4E, 4F, 4G, 4H and 8X and Supplemental Figures S4A, S5G and S5H using ANOVA, but the P values might.

Thank you for this point. We have now recalculated the indicated datasets using one-way ANOVA. As estimated by the referee, respective statistical significance did not change, except for Fig. 2M, but *P* values were mostly changed in the other panels, except for 2H. We still find no significance for panels 4F, 4G, and 4H. Panel 8X has been replaced by more detailed analysis (see point 2b).

For Figures S4A, S5G and S5H we chose the one sample t-test, as treatments are expressed as percentages, compared to control values, with a theoretical mean of 100.

Also, to meet Nature Communication requirements for displaying statistical analyses of small group sizes, bar diagrams in Fig. 2W, 8B, 9U, Suppl. Fig. 5G,H, Suppl. Fig. 9B have been changed to dot plots.

2a. LSP1 and Circularity. The first mention of the effect of LSP1 on circularity is in the presentation of Figure 8, in which LSP1 knockdown increases the circularity of macrophages that overexpress SV 1-830 GFP. At least, that was my reading of the text and the legend. The legend for Fig. 8X does not specify that the cells being scored were all SV 1-830 overexpressors, but that is the focus of this figure. The legend needs to be clarified.

The legend for Figure 8X was „Evaluation of circularity index for cells treated with control siRNA or LSP1-specific siRNAs, and overexpressing SV1-830-GFP”. Fig. 8X has been replaced by a more detailed analysis (Fig. 9M-U), with clear indication of SV1-830GFP overexpression in each case.

2b. Assuming that this interpretation is correct, the reader wants to know what effect LSP1 knockdown has on the circularity of macrophages that do NOT overexpress SV1-830. On the one hand, the cell with LSP1 siRNA in Figure 4 looks rounder than the cells with control siRNA in the same panel. On the other hand, LSP1 siRNAs increase cell speed (Fig. 2M) and cell invasion (Supplemental Figure S5). Both of these properties are generally associated with cell elongation. So, does LSP1 knockdown have opposite effects in untransfected macrophages versus macrophages expressing SV 1-830 GFP? If so, how does that fit with the hypothesis of dueling actomyosin contractile mechanisms? Data on the effect of LPS1 knockdown on circularity should be added to either Figure 2 or Figure 4, and any implications for the mechanism included in the Discussion.

For a more complete description of the LSP1-related phenotypes, we have now measured both circularity and aspect ratio (longest vs. shortest axis). Frequency distributions for both values are presented as the new Fig. 2P,Q. We are also including scatter plots that illustrate the interdependence of these values, with a graphical description of the respective cell shapes (Fig. 2R-V).

We find that LSP1 knockdown leads to a decrease of circularity and also to an increase of aspect ratio, indicating a more elongated cell shape. A regression analysis included in the new scatter plots also shows that, while both values are usually correlated in control cells (decreased roundness being linked to increased elongation), values for LSP1 knockdown cells are not as closely correlated. These observed effects are most probably the result of the formation of multiple protrusions that are associated with increased podosome cluster mobility (Fig. 2A-C). In addition, a bar diagram illustrates the most striking effects of LSP1 knockdown on aspect ratio ($AR \geq 1.3$) and circularity ($C \leq 0.8$) are included as the new Fig. 2W. We are also including an evaluation of podosome distribution in LSP1 knockdown cells (Fig. 2O) that supports the previous Fig. 2A-I.

For better comparability, we have now also performed these analyses for LSP1 knockdown cells that overexpress SV1-830. Former Fig. 8X has been replaced with the more extensive analyses, which forms the new Fig. 9M-U. Collectively, these results show that supervillin overexpression leads to a rescue of the morphological aberrancies observed upon LSP1 knockdown, probably due to the loss of competition for beta-actin binding in LSP1 knockdown cells and a thus more cortical localization of supervillin.

These new data are now also included in the Results (pp.6,7,12) and Discussion sections (p.16). As a further result of the restructuring, the panels of former Fig. 8A-I have now been moved to the Supplementary Material (Suppl. Fig. 9A-I). We are now also citing (Lomakin et al., NCB, 2015) for circularity and aspect ratio and discuss implications for LSP1-dependent actin bundling (p.15).

3. Figure 5 and LSP1 Co-immunoprecipitation with non-muscle myosin IIA (now Suppl. Fig. 7C and 7D). The demonstration that LSP1 binding to non-muscle myosin IIA requires the presence of

co-bound F-actin is an important point that is currently buried in the supplementary figures. Figure 5 has a lot of white space. I strongly recommend that panels C and D in Supplementary Figure 7 be incorporated into Figure 5 in the main text, probably as Fig. 5E and 5F.

We agree with the referee that this is an important point. The former Suppl. Fig. 7C and 7D have now been moved to the main part of the manuscript as the new Fig. 5D and 5E.

As a consequence of this rearrangement, free space became available in Suppl. Fig. 7. We thus moved Suppl. Fig. 8A-B (SV GST cosedimentation with actin isoforms + graph) to Suppl. Fig. 7C-D. Subsequently, Suppl. Fig. 9 then became the new Suppl. Fig. 8.

4. Text on pages 8 and 9 (lines 240 - 248), Figure 5 and Supplementary Figure 7. As currently presented the reader is asked to believe that the strongest evidence for a lack of direct binding of LSP1 to macrophage myosin IIA is the absence of co-sedimentation of LSP1 with rabbit skeletal muscle myosin II. In fact, their strongest evidence is the data with non-muscle myosin IIA (currently Suppl. Fig. 7C, discussed above). As the authors discuss for alpha-, beta- and gamma-actin, many proteins exhibit binding preferences for specific protein isoforms, and the myosin II heavy chains differ more than the actin isoforms. For instance, supervillin has been reported to bind only to non-muscle and smooth muscle myosin II proteins and not to rabbit skeletal muscle myosin II (Ref. 28), so the absence of binding of LSP1 to skeletal muscle in Suppl. Fig. 7B is not necessarily relevant to myosin II regulation in macrophages. I would recommend that the authors flip the presentation of the Suppl. Fig. 7C data (after moving it to Fig. 5) and the Suppl. Fig. 7B data in the text, making the less compelling data secondary to the more compelling evidence.

Thank you for this good suggestion. The text has been rearranged accordingly (pp. 8,9).

5. Figure 6. The title for Figure 6 is that "LSP1 and supervillin compete for myosin regulators in cells." That is a significant and important question. However, the data show only the change in localization of the interactors after over-expression of either GFP or SV 1-830 GFP. In a paper about LSP1, I would have expected Figure 6 to show shifts in the distributions of MLK, calmodulin and pMLC after over-expression of LSP1 or the LSP1 C-terminus, with the data in the current figure relegated to a supplementary figure. If there are two competing actomyosin-assembling machineries, I would expect to see complementary shifts after over-expression (or under-expression) of EITHER LSP1 OR supervillin. Assuming that to be true, including that data would greatly support the overarching hypothesis of this paper.

We appreciate the reviewer's suggestion. However, we do not think that these experiments would be able to support our data, as outlined in the following. The investigated myosin regulators (MLCK, calmodulin, pMLC) are already highly enriched in the cell periphery of control cells (Fig. 6C,K,S), as is LSP1 itself (Fig. 6B,J,R). Further overexpression of LSP1 is thus not expected to lead to enhanced enrichment of these regulators at the cell periphery. LSP1 knockdown, on the other hand, leads to a redistribution of supervillin towards the cell periphery (Fig. 8R,W). Interacting myosin regulators would thus also be kept at the cell periphery to a certain degree, even in the absence of LSP1. We thus believe that the redistribution of myosin regulators subsequently to overexpression of SV1-830 (Fig. 6) provides the clearest evidence for a competition between LSP1 and supervillin. We hope that the referee agrees with us on this point.

6. Figure 8. Similarly, the title for Figure 8 is "Differential subcellular recruitment of LSP1 and supervillin is based on the preferential binding of LSP1 to beta-actin." This is another important

question. Yet, the data show only the knockdown of cardiac ALPHA-actin and the results of expressing myosin IIA-binding sequences from supervillin. This data made sense when the prevailing hypothesis was preferential binding of supervillin to cardiac alpha-actin, but since the authors have shot down that hypothesis, this figure becomes an interruption of the logical flow, given the new biochemical results. The current data are supportive, but, as with Figure 6, the reader is primed to learn what happens after a knockdown of beta-actin with or without overexpression of LSP1 or the LSP1 C-terminus. For instance, does LSP1 or cardiac alpha-actin re-localize?

We have now also performed beta-actin knockdown experiments and analysed distribution and expression of alpha-actin and LSP1 in these cells. Interestingly, depletion of beta-actin did not result in discernibly altered distribution of LSP1, indicating additional mechanisms that stabilize LSP1 at the cell periphery. Still, the altered distribution of supervillin upon LSP1 knockdown clearly shows that preferential binding of beta-actin by LSP1 restricts the access of supervillin to this actin pool at the cell periphery. The new data are now shown in Fig. 8C-H and Suppl. Fig. 9A,B and are mentioned in the Results (p.12) and Discussion sections (pp.16,17).

To keep a logical flow of the manuscript, we have also moved the part describing the siRNA-mediated depletion of alpha-cardiac actin right before describing the beta-actin knockdown (pp.11,12). The respective panels of former Fig. 8A-I have moved to the Supplementary Material (new Suppl. Fig. 9). Importantly, the fixation and staining conditions especially for alpha-cardiac actin are not optimal for LSP1-specific antibodies (Fig. 8C-H). We have thus refrained from adding fluorescence intensity quantifications.

7. Figure 8A. A second reason for wanting to know about the results of a beta-actin knockdown is that one could argue that the knockdown of cardiac alpha-actin has had an effect due to a decrease in the total level of cellular actin, not necessarily because of a shift in the ratios of the actin isoforms. Opposite shifts in localization of relevant proteins after beta-actin knockdown, assuming that this doesn't also negatively affect alpha-actin levels, would greatly support the authors' thesis. This brings up the question of whether knockdown of one of the actin isoforms affects the levels of the other cellular actins. Figure 8A shows only a single immunoblot of alpha-cardiac actin and beta-actin after treatment with control or alpha-actin siRNA. The GAPDH loading control may be equivalent, but its overloading makes this assessment difficult to confirm. The numbers below GAPDH presumably refer to the relative levels of alpha-cardiac actin. Gamma-actin is not shown, and there are no quantifications for beta-actin. Levels of all actin isoforms should be quantified for both gamma and beta-actin knockdown.

We have now also established beta-actin knockdown and quantified levels of alpha-, beta, and gamma-actin, and also of LSP1, for both alpha- and beta-actin knockdown. We found that the effects of single knockdown experiments varied, and have thus performed each analysis 4 times, with respective quantifications. Treatment of macrophages with alpha-cardiac actin-specific siRNA led to a global ~ 1/3 reduction of alpha-cardiac actin levels and a wide range of reduction on the individual cell level. Neither beta-actin, gamma-actin or LSP1 levels were significantly changed in cell populations and also in individual knockdown cells. Treatment with beta-actin specific siRNA led to a reduction of ~60% of beta-actin levels, again with varying effects on the individual cell level. Also, while global levels of the remaining actin isoforms, and also of LSP1, were unchanged, individual beta-actin depleted cells frequently also showed reduced staining for alpha-cardiac actin. However, this effect was not significant in overall cell populations. These data are now included in the Results section (pp.11,12). Respective blots and quantifications are presented in the new Suppl.

Fig. 9A,B. As gamma-actin has not been part of the binding experiments performed earlier, the focus being on the beta- vs. alpha-actin dichotomy, a gamma actin knockdown was also not included in the new experiments. We hope the referee agrees with us on this.

Relatively minor points:

1. As the authors have learned, co-immunoprecipitated proteins cannot be said to "bind" to each other; the preferred phrase is "interact with" or "co-associate". Inappropriate "bind" or "binding" claims remain on lines 261 and 433. The biggest overstatement in this regard is the sentence on page 15, lines 427-428, that "these results showed that LSP1 acts as a hub for myosin activity by binding specific myosin regulators." This sentence needs to be deleted or corrected.

“Binding” has now been replaced with “interacts with”, and the sentence “These results showed that LSP1 acts as a hub for myosin activity by binding specific myosin regulators.” has been changed to “These results showed that LSP1 interacts with a variety of specific myosin regulators.”

2. Text on page 10, lines 305-306 versus Figure 8D. The text says that the SV 1-830 GFP signal in cells with reduced levels of cardiac alpha-actin is redistributed "away from podosomes and to the cell cortex." The figure shows a signal that does appear to be less central than the corresponding signal in Figure 6, in which the cells express endogenous levels of alpha-actin. However, it is still pretty far away from the "cortical" beta-actin and LSP1 signals in control cells. Perhaps "shifted towards the cell periphery" is more accurate than "cortical", unless the authors have better examples for this effect.

We agree with the referee. The phrase “away from podosomes and to the cell cortex” has now been replaced by “showed a shift ... towards the cell periphery”, as suggested.

3. Supplementary Figure 9. Why is Supplementary Figure 9 and the videos not presented in the Results section? No data should be presented for the first time in the penultimate paragraph of a paper's Discussion. In addition, these data set the stage for idea of cross-talking dynamic actomyosin machineries at the heart of this work.

We agree with the referee. Former Suppl. Fig. 9, now Suppl. Fig. 8 (see Major Point 3) is now mentioned in the Results section for the first time (p.11).

4. Text, pages 11-12, lines 342 - 345. Conversely, why are conclusions presented in the Results, instead of the Discussion? With the current data, most of these statements also are speculative. For instance, the conclusion that LSP1 and supervillin "compete for actin isoforms, especially beta-actin" is an inference from the actin binding data and SV 1-830 GFP re-organization after knockdown of alpha-actin. Also, the statement that "the cellular pattern of actin isoforms forms the basis for the differential distribution of BOTH (emphasis mine) [LSP1 and supervillin] in cells" is largely unsupported by the current data.

We agree that this paragraph should not be included in the Results section. It has now been removed. A part of this paragraph is now included in the Discussions section (p. 15).

5. Discussion, pages 14 - 15, lines 411 - 423. I'm surprised that the authors don't propose that the primary suggested mechanism for the shorter podosome lifetimes observed after LSP1 depletion is

due to the fact that supervillin is no longer excluded from binding to peripheral precursor podosomes, leading to podosome dissolution as described in this laboratory's previous paper on this subject. Given the theme of the paper, I would expect that the loss of LSP1-mediated recruitment of WASP and Arp2/3 would be prefaced by "Alternatively, or in addition".

Thank you for these good suggestions. Missing exclusion of supervillin from peripheral podosomes in the absence of LSP1 is now mentioned as a potential cause for the observed reduction in podosome lifetime. LSP1-mediated recruitment of WASP and Arp2/3 is now mentioned as a potential additional mechanism for the regulation of podosome lifetime (p. 14).

6. Discussion, page 17, lines 491 - 494. I am confused by the last clause in this sentence, "potentially leading to less coherence within the podosome array and thus giving rise to increased cluster formation." If the array is less coherent, then wouldn't the clusters be more dispersed?

In primary macrophage, podosomes usually form a single coherent array that is connected by myosin IIA-positive actin filaments that probably help to form the equidistant pattern by a tug-of-war mechanism. Formation of individual clusters is unusual, as it is based not on symmetric force distribution, but on several podosome-organizing centers. It thus points to a potential weakening (not necessarily a breakage) of at least some of these connections. Of course, other explanations may apply, but we just wanted to mention one likely possibility and to refrain from too much speculation.

Reviewer 2

I appreciate the substantial experimental work and the textual improvements that the authors added to the manuscript.

They adequately answered to my concerns.

I have no further points and do recommend the paper for publication.

Thank you for your constructive and positive evaluation of our paper.

REVIEWERS' COMMENTS:

Reviewer #1 (Remarks to the Author):

The authors have again been exceptionally responsive to the suggestions made in a second round of reviews, including the carrying out of additional experiments and a re-analysis of their statistical methods. The result is an impressive demonstration of the existence of two types of actomyosin arrays potentiated by LSP1 and supervillin. The point that this mechanism may be comparable to the situation in cells with two types of nonmuscle myosin II isoforms is especially intriguing, a point that might be worth including in the Abstract, assuming space can be found.

In my opinion, all of the changes that have been made are great. The one remaining issue I found is that the Methods section was apparently not updated to include the procedures employed for the new experiments. No information is available about how the beta-actin knockdown experiments were performed (source of siRNA, etc.). Neither is information included about the new measurements of cell circularity and aspect ratios. All details that others will need to reproduce these experiments need to be included.

Reviewer #1

1) The authors have again been exceptionally responsive to the suggestions made in a second round of reviews, including the carrying out of additional experiments and a re-analysis of their statistical methods. The result is an impressive demonstration of the existence of two types of actomyosin arrays potentiated by LSP1 and supervillin. The point that this mechanism may be comparable to the situation in cells with two types of nonmuscle myosin II isoforms is especially intriguing, a point that might be worth including in the Abstract, assuming space can be found.

*Thank you for the insightful and constructive review of our manuscript.
The abstract has now been modified accordingly, with the 150 word limit in mind.*

2) In my opinion, all of the changes that have been made are great. The one remaining issue I found is that the Methods section was apparently not updated to include the procedures employed for the new experiments. No information is available about how the beta-actin knockdown experiments were performed (source of siRNA, etc.). Neither is information included about the new measurements of cell circularity and aspect ratios. All details that others will need to reproduce these experiments need to be included.

The Methods section was indeed updated during the second revision to include the new experiments. Unfortunately, the changes were not highlighted in red. We apologize for this. We have now uploaded the most recent version including all new materials and procedures.